



# The diurnal stratocumulus-to-cumulus transition over land

Xabier Pedruzo-Bagazgoitia[1], Stephan R. de Roode[2], Bianca Adler[3], Karmen Babić[3], Cheikh Dione[4], Norbert Kalthoff[3], Fabienne Lohou[4], Marie Lothon[4], and Jordi Vilà-Guerau de Arellano[1]

[1]Meteorology and Air Quality Group, Wageningen University and Research, Wageningen, The Netherlands
[2]Delft University of Technology, Delft, Netherlands
[3]Institute of Meteorology and Climate Research, Karlsruhe Institute of Technology (KIT), Karlsruhe, Germany
[4]Laboratoire d'Aérologie, Université de Toulouse, CNRS, UPS, France

**Correspondence:** Xabier Pedruzo-Bagazgoitia (xabier.pedruzobagazgoitia@wur.nl)

**Abstract.** The misrepresentation of the diurnal cycle of boundary-layer clouds by large scale models strongly impacts the modeled regional energy balance in southern West Africa. In particular, recognizing the processes involved in the maintenance and transition of the nighttime stratocumulus to diurnal shallow cumulus over land remains a challenge. This is due to the fact that over vegetation, surface fluxes exhibit a much larger magnitude and variability than on the more researched marine
stratocumulus transitions. An improved understanding of the interactions between surface and atmosphere is thus necessary to improve its representation. To this end, the DACCIWA measurement campaign gathered a unique dataset of observations of the frequent stratocumulus to cumulus transition in southern West Africa. Inspired and constrained by these observations, we perform a series of numerical experiments using Large Eddy Simulation. The experiments include interactive radiation and surface schemes where we explicitly resolve, quantify and describe the physical processes driving such transition. Focusing
on the local processes, we quantify the transition in terms of dynamics, radiation, cloud properties, surface processes and the evolution of dynamically relevant layers such as subcloud layer, cloud layer and inversion layer. We further quantify the processes driving the stratocumulus thinning and the subsequent transition initiation by using a liquid water path budget. Finally, we study the impact of mean wind and wind shear at cloud top through two additional numerical experiments. We find that the sequence starts with a nighttime well-mixed layer from surface to cloud top, in terms of temperature and humidity, and
transitions to a prototypical convective boundary layer by the afternoon. We identify radiative cooling as the largest factor for the maintenance leading to a net thickening of the cloud layer of about $18 \text{ g m}^{-2} \text{ h}^{-1}$ before sunrise. Four hours after sunrise, the cloud layer decouples from the surface through a growing negative buoyancy flux at cloud base. After sunrise, the increasing impact of entrainment leads to a progressive thinning of the cloud layer. While the effect of wind on the stratocumulus layer during nighttime is limited, after sunrise we find shear at cloud top to have the largest impact: the local turbulence generated
by shear enhances the boundary layer growth and entrainment aided by the increased surface fluxes. As a consequence wind shear at cloud top accelerates the breakup and transition by about 2 hours. The quantification of the transition and its driving factors presented here sets the path for an improved representation by larger scale models.





## 1 Introduction

Stratocumulus (Sc) clouds play a critical role on the radiative balance of the planet given their high albedo (Hartmann et al., 1992; Chen et al., 2000) and extensive cover worldwide (Eastman and Warren, 2014; Eastman et al., 2014). These boundary-layer clouds are a common feature in the southern West Africa (SWA), and recur in the night and morning during the Monsoon

season between May and September (van der Linden et al., 2015; Hill et al., 2018). Possible future changes in highly sensitive Sc forcings in SWA, such as anthropogenic regional aerosol increase (Boucher et al., 2013) or the global $CO_2$ rise (Schneider et al., 2019), further motivate a better understanding of the boundary-layer cloud dynamics over land in SWA.

During the monsoon season the intertropical convergence zone shifts northward till $15°$ N, facilitating the extension of the maritime air masses inland. The arrival of the cooler, but not necessarily moister, mass of air more than a 100 km inland

facilitates the onset of Sc clouds over land (Adler et al., 2019; Babić et al., 2019; Dione et al., 2019). The fact that this mass of air is characterized by cloudless conditions when over the sea reveals the importance of the land and other local factors on the cloud formation and maintenance (Adler et al., 2019; Babić et al., 2019; Lohou et al., 2019). Lohou et al. (2019) extended the previous work and summarized the four phases leading from cloud formation to dissipation. In addition, they described three observed scenarios for the breakup and dissipation of the Sc deck along the day. Such scenarios differed on the Sc coupling to

surface and on the presence of convective clouds below the Sc.

The high albedo of low Sc clouds and its underestimation by most climate models lead to significant biases on the regional surface energy balance if the evolution and spatial structure of the cloud field is not correctly represented (Hannak et al., 2017). More specifically, the maintenance, dissipation or transition to other cloud forms of the Sc cloud layer after sunrise has strong implications in the regional energy balance (Knippertz et al., 2011; Hannak et al., 2017; Lohou et al., 2019). To improve our

understanding and better quantify the effects of Sc clouds over land in a observation-scarce region, the Dynamics-aerosol-chemistry-cloud interactions in West Africa (DACCIWA) project deployed an extensive network of observations during June and July in 2016 comprising three fully instrumented supersites (Knippertz et al., 2015; Flamant et al., 2018; Kalthoff et al., 2018). The resultant dataset of high spatio-temporal observations of the cloud transition allows us to tackle two important questions. Firstly, it allows us to understand the transition (Lohou et al., 2019) and, using idealized numerical simulations,

reproduce a characteristic stratocumulus to cumulus (Sc-Cu) transition with typical conditions of SWA. Secondly, we aim at identifying the physical processes and their complex interplay that leads to a breaking up of the Sc deck.

Here, we extend on the impacts of the land-atmosphere interactions on the Sc-Cu transition and breakup. Previous studies have largely relied on high resolution explicit modeling, e.g. Large Eddy Simulation (LES), of marine Sc clouds. Over sea, surface fluxes are low and show little diurnal variation. Evaporation from the sea provides the necessary moisture to maintain

the Sc layer, that is well-mixed down to the surface by the turbulence generated at the cloud top by radiative cooling (Wood, 2012). Transitions from Sc to shallow cumulus have also been studied through LES mostly in maritime conditions (Bretherton et al., 1999a; Sandu and Stevens, 2011; de Roode et al., 2016). Such transitions are typically investigated using a Lagrangian approach in which the trajectory of an air mass is followed as it is advected from the subtropics towards the equator. An increasing sea surface temperature and decreasing subsidence is usually imposed along the trajectory, leading to increasing



latent heat fluxes and boundary layer height, respectively. The main mechanism for such transitions over sea is the increase in buoyancy along the cloud layer by higher latent heat flux, leading to larger entrainment velocities aided by the subsidence decrease, and the eventual dissipation of the Sc cloud layer. Over land, however, such transitions may have different drivers, given their differently partitioned surface fluxes as well as their larger magnitude and diurnal variability than over sea. Ghonima

et al. (2016) performed a thorough idealized LES study on Sc-Cu transitions both over land and over sea. They based all their cases on vertical profiles of mid-latitude marine conditions and prescribed different Bowen ratios to regulate the surface fluxes over land. In contrast, atmospheric conditions in near-equatorial SWA are characterized by a moister and warmer atmosphere as well as stronger solar irradiation and, locally, larger evapotranspiration. These differences pose the question of whether the mechanisms and physical processes identified by Ghonima et al. (2016) remain relevant for SWA. Our study thus aims

at filling the knowledge gap on turbulence resolving numerical experiments of Sc-Cu transitions taking place over land and, specifically, in sub-tropical atmospheric conditions. We systematically focus on the following processes and the rol played in the maintenance of the Sc and its transition to cumulus clouds: radiation, entrainment and the land surface fluxes. Radiation is the source for cloud maintenance during night and, as the day evolves, a factor for dissipation. Entrainment is known to affect the cloud layer by drying and warming it, rising it and weakening the thermal inversion. The land surface fluxes respond to

variations in wind and radiation and affect the transport of heat and moisture to the cloud layer as well as the entrainment. In addition, we briefly study the evolution of metrics frequently used by parameterizations in larger scale models along the Sc-Cu transition .

Finally, during the DACCIWA campaign a recurrent low level jet along the cloud layer was observed (Adler et al., 2019; Dione et al., 2019), raising an additional question on the effects of wind shear on Sc and its transition d(Lohou et al., 2019).

Previous work on modeled sheared Sc over sea suggests that shear at cloud top lowers the water content of Sc by enhancing the entrainment rate (Wang et al., 2008, 2012). Mechem et al. (2010) presented a land Sc case and briefly studied the effects of shear. They similarly concluded that entrainment velocity and cloud liquid water content decreases in presence of cloud-top wind shear. However, to the best of our knowledge, there is no work studying the effects of wind shear on stratocumulus to cumulus transitions. Thus, we additionally perform some sensitivity studies on the effect of wind and wind shear at cloud top

on the Sc-Cu transition.

Our research seeks to answer these three research questions:

- How is a stratocumulus to cumulus transition over land characterized? What is the relevance of the local processes?

- How do the contributions of each physical process vary with time during the maintenance, thinning and transition of the cloud layer?

- How do metrics relevant for larger scale models quantify the transition?



## 2  Methods

### 2.1  Dutch Atmospheric Large Eddy simulation (DALES)

To explicitly resolve the Sc-Cu transition we perform our numerical experiments using the Dutch Atmospheric Large Eddy Simulation (DALES) (Heus et al., 2010; Ouwersloot et al., 2016). LES models explicitly resolve most of the energy-containing turbulent motions in the boundary layer, including the stratocumulus and shallow cumulus cloud dynamics. Based on the initial work of Nieuwstadt and Brost (1986), DALES is a LES model that has been used individually or within a model intercomparison for a broad range of cases, from clear sky diurnal cycles (Pino et al., 2003) to boundary layers topped by stratocumulus (Blossey et al., 2013; van der Dussen et al., 2015) or cumulus (Siebesma et al., 2003; Vilà-Guerau de Arellano et al., 2014), including Sc-Cu transitions over sea (van der Dussen et al., 2013; de Roode et al., 2016). Here, we use the DALES 4.1 version. We describe below the relevant parameterizations for this study.

- An interactive land surface model with a mechanistic representation of plant behavior. It regulates the surface latent, and sensible heat fluxes, as well as the $CO_2$ flux depending on environmental variables such as $CO_2$ concentration, atmospheric vapor pressure deficit, temperature, soil moisture and surface wind (Jacobs and de Bruin, 1997; van Heerwaarden et al., 2010; Vilà-Guerau de Arellano et al., 2015). It is upgraded with a 2 big-leaf (sunlit and shaded leaves) scheme allowing for different penetration rates of direct and diffuse radiation along the canopy (Pedruzo-Bagazgoitia et al., 2017). The fact that surface fluxes are higher and more variable over land, responding to environmental variables and potentially altering the boundary layer and cloud structure, explains the need for an interactive scheme as the one presented here.

- The two-stream radiation scheme RRTMG (Iacono et al., 2008). It is used to provide longwave and direct and diffuse shortwave radiation at each gridbox dependent on liquid water and other chemical compounds.

- The microphysics scheme by Khairoutdinov and Kogan (2000), specifically designed for Sc clouds. It includes cloud-droplet sedimentation, found to be highly relevant in the representation of Sc clouds (Ackerman et al., 2004; Bretherton et al., 2007; Dearden et al., 2018).

### 2.2  Observations

We base our idealized study on observations taken during the field campaign of the DACCIWA project during the months of June and July 2016. We focus on the observations of 26th June 2016 at the Savé supersite. On this day a stratocumulus deck was observed during the night and morning above Savé, followed by a cloud base rise and breakup during the late morning and afternoon (Dione et al., 2019). We briefly describe below the methods and observations used to inspire our idealized study. For a fully detailed explanation of the observations and the dataset, the reader is referred to Kalthoff et al. (2018) and Bessardon et al. (2019), respectively.





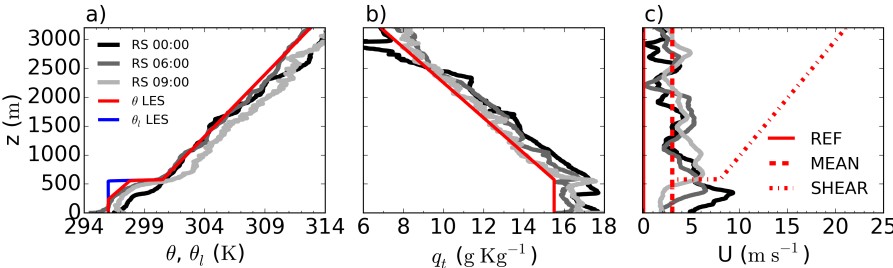

**Figure 1.** Vertical profiles of potential temperature (a), total specific humidity (b) and wind (c) as observed through radiosondes (at 00 00, 06 00 and 09 00 UTC from black to light grey) on 26th June 2016 and as prescribed in the idealized LES experiments (red and blue). The three experiments REF, MEAN and SHEAR differ only in the prescribed wind profiles.

– Radiosondes were performed with the MODEM radiosounding system. The temperature and relative humidity of the air were measured with a 1 second temporal resolution ($\simeq 4-5$ m in vertical resolution). The wind speed, direction and the pressure were determined based on the radiosonde GPS coordinates (Derrien et al., 2016).

– The cloud base height is measured by a continuously running ceilometer measuring backscatter profiles with a 1 minute resolution. From backscatter profiles a cloud base height is obtained using the manufacturer algorithm. The data is available at Handwerker et al. (2016).

– The cloud top height is measured by a dual-polarized cloud radar, which allows to distinguish between hydrometeors and other targets. The cloud top height is estimated from the 5 min averaged reflectivity profiles of hydrometeors applying a threshold of -35 dBz (Bauer-Pfundstein and Goersdorf, 2007). Therefore, reflectivities larger than -35 dBz are considered clouds. The data is available at Handwerker et al. (2016).

– The cloud cover is calculated as the percentage of cloud base height measurements below 1000 m (Adler et al., 2019; Zouzoua, 2019). The values are averaged over 19 days during the campaign to prevent too high variability by single point observations of individual days.

– The surface fluxes are obtained from an energy balance station deployed over a mix of grass and bushes. The 30-min turbulent fluxes are calculated from high-frequency (20 Hz sampling rate) ultra sonic anemometer measurements of wind speed and sonic temperature and humidity measurements from fast infrared hygrometer (LICOR) by applying eddy-correlation method (Mauder et al., 2013). The data is available at Kohler et al. (2016).

– The two sets of turbulent kinetic energy measurements are calculated from the anemometer measurements of wind speed at $4$ m and $7.8$ m by two energy balance stations deployed over a mix of grass and bushes and over corn, respectively.



## 2.3 Model settings and initial conditions

Constrained by the surface and upper atmospheric observations, we design an academic case to be simulated through LES. Our aim is, by means of an idealized numerical experiment, to simulate a Sc-Cu transition, including the Sc breakup, during typical atmospheric conditions in SWA rather than the reproduction of an exact day occurred during the DACCIWA measurement campaign. In particular, we study the Sc-Cu transition of a coupled case as described by Lohou et al. (2019).

We design a 12x12 km$^2$ wide and 3.2 km high domain, with a gridbox size of 50x50x4 m$^3$ resulting in 800 vertical levels. Such high vertical resolution is required in order to reduce the overestimation of mixing and entrainment typical of coarser LES simulations with Sc (Bretherton et al., 1999b; Stevens et al., 2005). Although at this vertical resolution processes such as evaporative cooling and cloud top mixing might still be overestimated (Stevens et al., 2005; Mellado, 2017), a much finer resolution, or a Direct Numerical Simulation approach, would not allow computationally for an integrated simulation of both cloud top and surface. As it will be shown later, both interfaces play a critical role in the development and transition studied here. We use periodic boundary conditions on the horizontal directions. We start the experiment at 3 30 UTC to allow for one hour of spin up of the Sc layer and end the experiment at 18 30 UTC after sunset.

We prescribe a subsidence profile, following Bellon and Stevens (2012), of the shape $w_{subs}(z) = -w_0(1 - e^{\frac{-z}{z_w}})$, with $w_0 = 5.3$ mm s$^{-1}$ and $z_w = 300$ m. Such a profile translates to $w_{subs} = -4.51$ mm s$^{-1}$ at the initial cloud top height of 570 m. Our choice for the subsidence profile is justified given the uncertainty and high temporal variability in subsidence profiles, as well as its large spread among regional simulations carried out with the Consortium for Small-Scale Modeling (COSMO) within the DACCIWA project or ERA-interim reanalysis. To limit the complexity of our idealized experiments and focus on the interaction of the surface and boundary-layer processes, we prescribe no advection of heat or moisture at any height. Adler et al. (2019) and Babić et al. (2019) found cold air advection necessary for the formation of the cloud layer. Yet its relevance decreased as sunrise approached, thus justifying our assumption during our time of interest.

For all the experiments we calculate the vertical profiles of the radiative fluxes every minute. In doing so, we quantify how radiative fluxes are perturbed by the liquid water related to cloud dynamics and how they interact with the surface. This is done to account for fast fluctuations of net radiation at cloud top and surface. The latter is relevant given its potential to alter surface fluxes and, thus, the evolution of the boundary layer and clouds (Vilà-Guerau de Arellano et al., 2014; Gronemeier et al., 2016; Sikma and Vilà-Guerau de Arellano, 2019). Based on aircraft observations during the DACCIWA campaign (Taylor et al., 2019), the cloud droplet number concentration is set to 300 cm$^{-3}$ and remains constant throughout the experiment.

We show in Fig.1 the vertical profiles obtained through three radiosondes during the night and morning of 26$^{th}$ June 2016. The radiosonde at 6 00 UTC, the closest to our initialization time, shows a strong increase in potential temperature of about 3 K at 570 meters high. Above, all radiosondes show similar temperature lapse rates of about 4.6 K km$^{-1}$. Subtropical marine Sc clouds are frequently capped by a strong drying above cloud top (Duynkerke et al., 2004; Wood, 2012). Yet none of the radiosonde profiles show any strong drying above 570 meters. If any, they show a humidity increase above cloud layer. Such increase could be related to the previously questioned reliability of radiosonde measurements as they exit the cloud layer through their ascension (Lorenc et al., 1996; Mechem et al., 2010; Babić et al., 2019). Situations without a dry jump above




Sc cloud top have been previously reported over land (Mechem et al., 2010) and are more typical of arctic climates (Morrison et al., 2012).

The observations demonstrate that the idealized experiment's initial conditions lie within typical meteorological conditions in SWA. The initial idealized profiles prescribe a well mixed layer up to 570 meters with liquid-water potential temperature $\theta_l = 296$ K and specific humidity $q_t = 15.5$ g kg$^{-1}$. Such thermodynamic conditions result in a domain-covering cloud layer from 226 m to 570 m high, topped by a jump of $4.5$ K in temperature, but without a jump in specific humidity. Above 570 m the potential temperature and total moisture idealized profiles exhibit constant slopes of $4.67$ K km$^{-1}$ and $3.29$ g kg$^{-1}$km$^{-1}$ respectively. Given the spread in vertical profiles by radiosondes, we performed additional simulations exploring variations in the profiles of $0.5$ K and $0.5$ g kg$^{-1}$. Results showed very similar development of the Sc-Cu transition.

Our reference experiment REF prescribes no wind at all heights. To study the effect of wind and wind shear, we perform two additional numerical experiments, MEAN and SHEAR, where we account for different idealized wind effects. This sensitivity analysis is motivated by the recurrent winds with the shape of low-level jet, such as those in Fig. 1c, that were frequently observed during the DACCIWA campaign (Kalthoff et al., 2018; Adler et al., 2019; Dione et al., 2019). Failed attempts to maintain a low level jet-like wind profile together with the Sc cloud layer in preliminary experiments suggest that the jet-like wind is the result of large scale dynamics, and, thus, beyond the scope of the present study on local factors. The large scale origin of the low level wind is also supported by more detailed observational analysis (Babić et al., 2019; Adler et al., 2019; Dione et al., 2019). Following our idealized approach, the initial wind speed and wind direction are inspired by the observations and adapted to better study how these effects influence the Sc-Cu transition. In this case, the mean wind and the shear at the cloud top are considered. We prescribe a constant horizontal wind of 3 m s$^{-1}$ along the whole vertical profile in MEAN based on above cloud layer radiosonde observations (Fig.1c). Consistent with our idealized setting, we assume the wind to blow only along the x-direction and without prescribed directional shifts with height. In SHEAR we add a jump of 5 m s$^{-1}$ to the mean 3 m s$^{-1}$ at cloud top to represent a wind shear of similar magnitude as the observed low level jet. The values prescribed here for the simplified effects of the low level jet are representative not only of the day studied here but also of the whole measurement campaign (Dione et al., 2019). The free troposphere wind shows a constant increase of 5 m s$^{-1}$km$^{-1}$ in SHEAR. Our aim here is to maintain a shear contribution as the cloud layer rises. We prescribe geostrophic winds identical to the initial wind profiles, as the goal is to observe the impact of wind on the transition and not vice versa. In summary, differences between MEAN and REF serve in identifying the role played by a mean wind, which will mainly enhance the surface fluxes. MEAN and SHEAR differences show the impact of the local shear at cloud top.

## 3  Results

### 3.1  Evolution of the transition

Figure 2 shows the diurnal evolution of cloud height, cover, and liquid water path (LWP) and their connection with surface turbulent fluxes in the REF simulation. It also includes the observations corresponding to the day by which our case is inspired. At Fig. 2a both cloud top and base remain approximately constant for the first hours. The LWP values are on the high end of





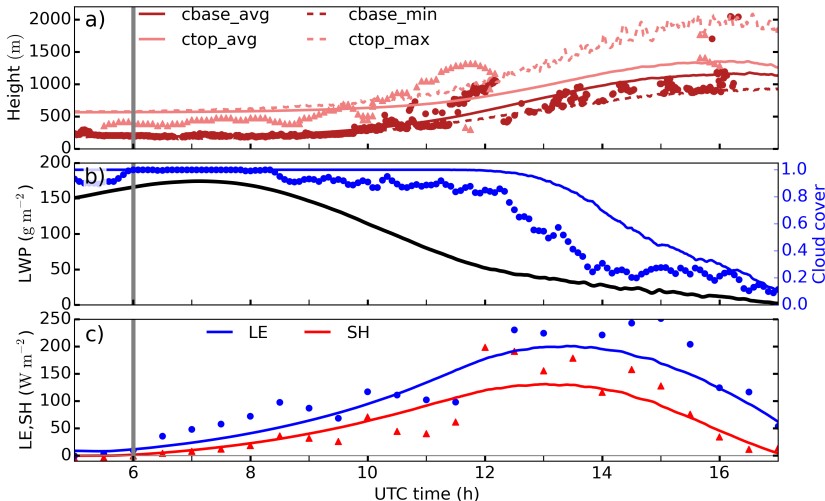

**Figure 2.** Time series of the domain average cloud base (cbase_avg in full dark red line) and cloud top (ctop_avg in full light red line), maximum cloud top (ctop_max in dashed light red line) and minimum cloud base (cbase_min in dashed dark red line) (a), liquid water path and cloud cover (b) and latent and sensible heat fluxes (c) in REF. Observed cloud base and cloud top heights on 26th June are represented by dark red circles and light red triangles, respectively, in (a). Observed cloud cover, averaged over 19 campaign days is shown in blue circles in (b). Observed latent and sensible heat fluxes are shown by blue circles and red circles, respectively, in (c). The vertical grey lines indicate the sunrise time.

domain average LWP for marine stratocumulus cloud (Wood, 2012) and coincide with observed ones during the DACCIWA campaign (Babić et al., 2019; Kalthoff et al., 2018). The initially constant cloud top height coincides with the boundary layer height. As a result, the boundary layer height evolution can be expressed using mixed layer theory by the relation that equates the entrainment velocity and the subsidence. It reads, assuming horizontally homogeneous conditions (Lilly, 1968):

$$\frac{\partial h}{\partial t} = w_e + w_{subs}(h) \simeq 0 \tag{1}$$

with $h$ the boundary layer height defined as the height of minimum buoyancy flux, $w_e$ the entrainment velocity and $w_{subs}(h)$ the subsidence, depending on height as described in Sec.2.3, at $h$. For this experiment and before sunrise $w_e \simeq -w_{subs}(h) = 0.45$ cm s$^{-1}$, which is in the same order of magnitude as previously reported nocturnal marine Sc cases (Stevens et al., 2003). Between 2 to 3 hours after sunrise (6 00 UTC) the cloud layer begins to rise and subsequently decreases its water content. Along this time the domain average cloud base cbase_avg follows the observed cloud base, and so do the surface fluxes with the observed ones. The onset of the convective phase, defined by Lohou et al. (2019) as the time when the sensible heat flux $SH > 10$ W m$^{-2}$, takes place between 7 00 and 7 30 UTC according to observations and at 6 55 UTC in REF. The breakup in the cloud layer, defined as the time when cloud cover ($cc$) is below 1, takes place at around 11 30 UTC in our experiment and coincides with the observed sharp increase in surface fluxes of about 150 W m$^{-2}$, i.e., a threefold increase compared





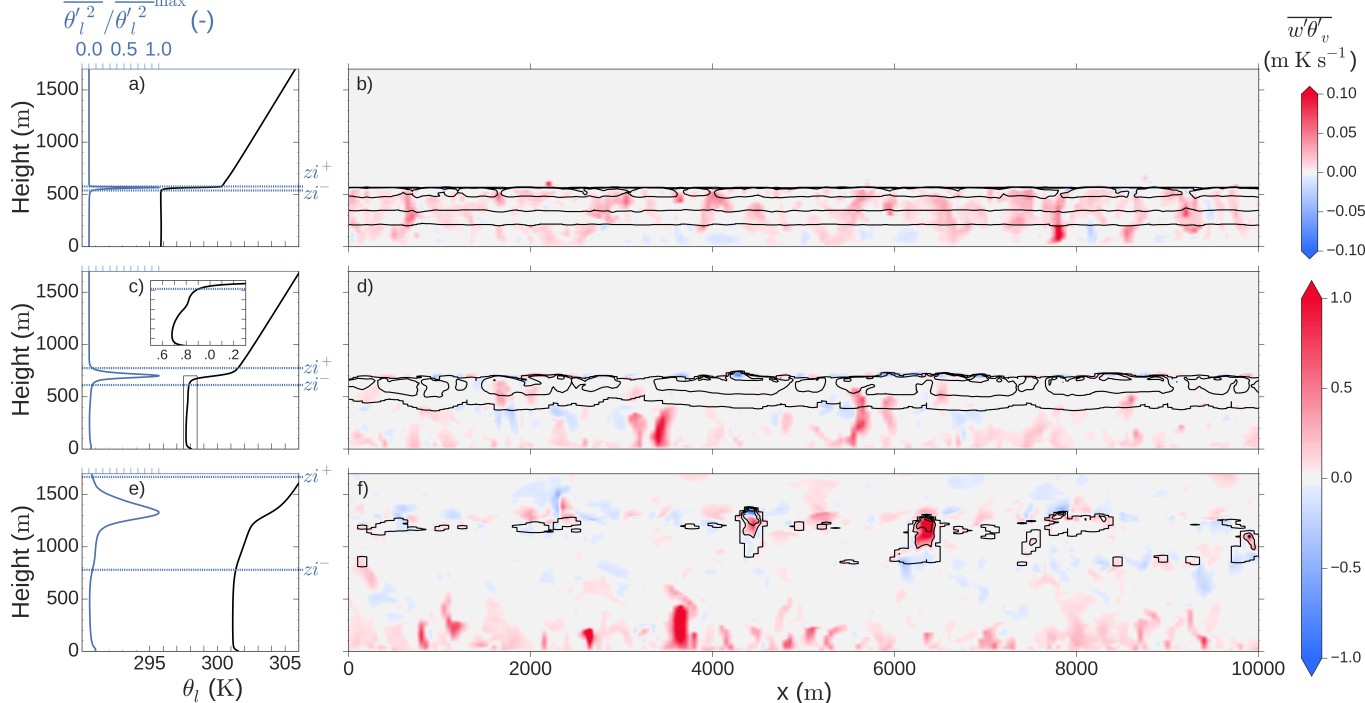

**Figure 3.** On the left, slab averaged vertical profile of liquid potential temperature $\theta_l$ (black) and $\overline{\theta_l'^2}$ normalized over its maximum value (blue). The inversion layer upper $zi^+$ and lower $zi^-$ limits are indicated in dashed blue horizontal lines. On the right, horizontal cross-section of buoyancy flux $\overline{w'\theta_v'}$ (in colours, red (blue) indicating upwards (downwards) movement of buoyantly positive (negative) air) and cloud liquid water (in black contour lines every 0.3 $g_w$ $Kg_a^{-1}$). Top plot corresponds to 5 00 UTC, center to 11 00 UTC and bottom to 14 30 UTC. The inset at c) is an expanded version of the rectangle in the same subfigure.

to before-breakup values. This sudden change reveals that surface fluxes are radiation-driven. The good agreement in the surface flux partitioning as modeled and observed justifies the use of a land surface model sensitive to several environmental variables at surface (see Sec. 2.1). After 11 30 UTC observations show large variability in measured cloud base heights (Fig. 2a), suggesting either the presence of shallow clouds below the Sc cloud layer or the breakup of Sc layer (Lohou et al., 2019).

5 After cloud break up, $cc$ decreases quasi-linearly until the end of the simulation. The same pattern for cloud cover is shown by observations, although one hour in advance. Note, however, that the observations of $cc$ are averaged over 19 days timeseries selected due to a cloud onset before 4 00 UTC (Zouzoua, 2019). The variability between the days considered in the average also explains the $cc$ values below 1 before 6 00 UTC and after 8 30 UTC.

### 3.2 Transition on turbulence and radiation

10 The turbulent spatial structure explaining this transition from typical nocturnal stratocumulus to convective clouds is shown in Figure 3 through the buoyancy flux and temperature profiles. The initial stages of the LES experiment (Fig. 3a,b) present a





well mixed and fully coupled layer from surface to cloud top. This layer is limited by a strong jump in liquid water potential temperature $\theta_l$ of about 4 K at 5 00 UTC (Fig. 3 a) at around 550 m, and a very narrow inversion layer. We quantify this layer through their lower and upper limits $zi^-$ and $zi^+$, respectively. These heights are defined, following van der Dussen et al. (2016), as the heights above and below, respectively, the maximum in slab averaged $\overline{\theta_l'^2}$, $\overline{\theta_l'^2}^{max}$, at which 5% of $\overline{\theta_l'^2}^{max}$ is

reached. After some hours the boundary layer evolves to a well mixed subcloud layer with a conditionally unstable cloud layer aloft at 14 30 UTC and a very broad inversion layer. Such evolution of the inversion layer allows us to interpret the typically conditionally unstable region of the cloudy layer in convective conditions as an expanded analogue of the very sharp inversion layer in Sc clouds. Thus, to correctly represent the transition studied here it is necessary to treat the evolution as a transition where the inversion layer expands as the boundary layer grows. A more detailed evolution of the inversion layer is given in

Fig. 5.

In the absence of mechanical production of turbulence, buoyancy is the only driving mechanism for turbulence. Figs 3 b,d,f quantify the shift of buoyancy-driven turbulence generation from cloud top radiative cooling at 5 00 UTC to surface warming at 14 30 UTC. Note the change in scale by a factor of 10 in $\overline{w'\theta_v'}$ between Fig 3b and 3d. Such a difference in magnitude shows that the surface-driven turbulence after sunrise becomes stronger, about 10 times, than the one created by cloud-top cooling. In

fact, the cloud-top cooling contribution to the buoyancy flux is in part diminished by a compensating condensational warming within the cloud layer. At 11 00 UTC there is a critical moment in the transition: the cloud layer remains rather homogeneous, but the mixed layer is now simultaneously driven both by surface warming and cloud top cooling. As it will be shown later (Figs. 4 and 6), the penetration of shortwave radiation through the cloud layer down to the surface is key in regulating both phenomena. The warming of the cloud layer leads to a decoupling of the cloud and subcloud layers. This is already visible at

11 00 UTC with a temperature difference between layers of about 0.2 K at 400 m high (see inset in Fig. 3c).

By resolving interactively the radiation transfer along the cloud layer and the surface response we gain insight on the dynamical transition, as shown in Figure 4. There, we observe how the vertical velocity distribution at the middle of the boundary layer starts from a situation with limited extreme velocities (between $-1.3$ and 1 m s$^{-1}$) and a negative skewness of $S_w = -0.3$ at 5 00 UTC, where $S_w = \frac{\overline{w'^3}}{\overline{w'^2}^{\frac{3}{2}}}$. This value for $S_w$ lies within the limits of typical marine Sc clouds (Ghate

et al., 2014). It then evolves to a prototypical convective-boundary-layer (CBL) skewed distribution with a larger spread of vertical velocities at 11 00 UTC (between $-1.5$ and 2.7 m s$^{-1}$) and $S_w = 1.2$ at half of the boundary layer height, having skewness values typical of dry convective boundary layers (Lenschow et al., 2012) or situations with cumulus coupled to Sc clouds (de Roode and Duynkerke, 1996). Similar values for $S_w$ are found at 14 30 UTC, with minimum and maximum vertical velocities between $-1.8$ and 3.5 m s$^{-1}$, respectively. The transition from stratocumulus to prototypical convective conditions

is reinforced by the evolution of the radiative profiles. Figure 4d shows an initial net radiative divergence at the cloud top of 43 W m$^{-2}$. The related cooling drives the mixed layer at 5 00 UTC. At this time the radiative cooling is stronger than the warming by entrainment as the mixed layer cools at a rate of about 0.1 K h$^{-1}$ before sunrise (not shown). By 11 00 UTC there is a net radiative warming along the cloud layer (between 400 and 650 m high, see Fig. 3) due to the absorption of shortwave radiation within the cloud layer. Shortwave radiation locally warms up to 1.1 K h$^{-1}$ the lower two thirds of the cloud layer





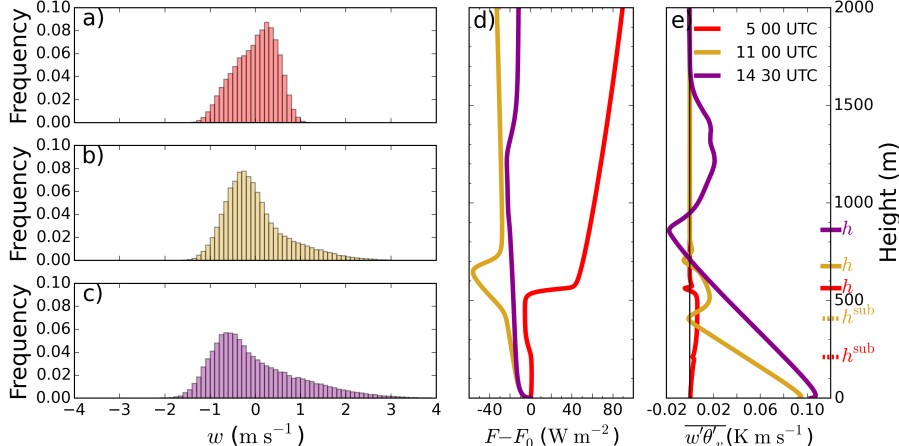

**Figure 4.** On the left, frequency distribution of vertical velocities at $\frac{h}{2}$ at 5 00 UTC (top), 11 00 UTC (center) and 14 30 UTC (bottom). On the center (d), vertical profile of slab net radiative flux normalized over the surface value at 5 00 UTC (red), 11 00 UTC (dark yellow) and 14 30 UTC (purple). On the right (e) and following the same color code, slab averaged buoyancy flux $\overline{w'\theta'_v}$. The subcloud layer height $h^{\mathrm{sub}}$ and the boundary layer height $h$ are shown for each time at the right vertical axis in (e). At 14 30 UTC both heights coincide.

due to the 44 W m$^{-2}$ of absorbed shortwave radiation along its travel through the cloud layer (not shown). The high cloud droplet number, 300 cm$^{-3}$, is likely to influence positively such net warming.

This net radiative warming along the cloud layer reinforces the warming driven by entrainment of free tropospheric air. The combination of both processes is critical for the decoupling of the cloud and subcloud layers. As it will be shown later (Fig.

6), it also plays a role on the thinning of the Sc and the reduction of turbulence generation at cloud top. Figure 4e shows the profile of the buoyancy flux, closely linked to the role of radiation. The averaged buoyancy flux shows a similar transition starting from prototypical nocturnal Sc clouds at 5 00 UTC, with positive buoyancy along the whole layer up to 550 m and a local minimum at cloud base due to latent heat release (Bretherton and Wyant, 1997; Wood, 2012). We define the height of such minimum as the subcloud layer height $h^{\mathrm{sub}}$. The definition of $h^{\mathrm{sub}}$ is necessary to better quantify the decoupling of

the stratocumulus layer from the surface, as it will be shown in Fig. 10. At cloud top Fig. 4e presents an absolute minimum at the boundary layer height $h$. It then evolves to profiles common in cumulus topped convective boundary layers (Siebesma et al., 2003) or decoupled Sc cloud layers (Wood, 2012) with linearly decreasing $\overline{w'\theta'_v}$ up to the cloud base, and buoyantly active convective clouds above 950 m. Note that under such conditions the boundary layer height and the subcloud top height coincide and $h^{\mathrm{sub}} = h$. The buoyancy flux profile at 11 00 UTC shows the decoupling of the cloud layer from the surface by

the enhancement of the local minimum at $h^{\mathrm{sub}}$ at 400 m. This nearly negative value in the vicinity of the cloud base has already been described as an indication of decoupling and hampered transport of moisture (and heat, in our case) from the surface to the cloud layer (Turton and Nicholls, 1987; Stevens, 2000; Lewellen and Lewellen, 2002). This will be further explored in Fig. 10.



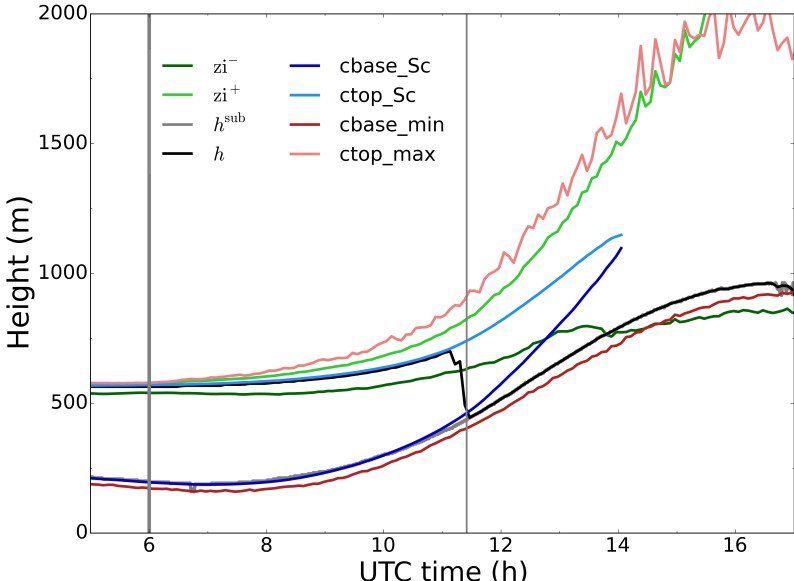

**Figure 5.** Time series of inversion layer top $zi^+$ (light green) and bottom $zi^-$ (dark green) heights, boundary layer height $h$ (black) and subcloud buoyancy minimum height $h^{\text{sub}}$ (grey), stratocumulus cloud base cbase_Sc (dark blue) and top ctop_Sc (light blue) heights, and minimum cloud base cbase_min (dark red) and maximum cloud top ctop_max (light red) heights. Sunrise time and cloud breakup time are indicated by the thick and thin grey lines, respectively.

We show in the timeseries in Fig. 5 the evolution of the variables that better reflect the dynamics of the Sc-Cu transition: we show the inversion layer upper and lower limits $zi^+$ and $zi^-$, respectively, the subcloud top height $h^{\text{sub}}$ and boundary layer height $h$ shown in Fig. 4, the maximum cloud top and minimum cloud base heights as in Fig. 2, and we additionally calculate the Sc cloud base cbase_Sc and cloud top ctop_Sc. These are defined as the height of the lowest and highest vertical level, respectively, with a slab averaged cloud fraction higher than $40\%$. After 10 00 UTC the Sc cloud base rises faster than the minimum cloud base. This is analogous to the slower rise of Sc cloud top compared to the maximum cloud top. Due to a faster rise of Sc cloud base than Sc cloud top, there is a thinning of the Sc layer eventually dissipating at 14 00 UTC. Lohou et al. (2019) observed a similar cloud thinning pattern based solely on observations. The cloud and subcloud layer dynamics divert from coupled Sc conditions, i.e. well mixed layer from surface to cloud top, several hours before, as it was shown in Figs. 3 and 4. Between 11 00 and 11 30 UTC, i.e. before the cloud breakup, $h$ shifts from the cloud top to the subcloud layer top represented by $h^{\text{sub}}$. The evolution of the inversion layer, indicated by $zi^+$ and $zi^-$, reveals a broadening of the inversion layer from a very thin layer ($\sim 50\,\text{m}$) across cloud top during the first few hours to a region thicker than $1\,\text{km}$ in the afternoon due to the cumulus clouds.





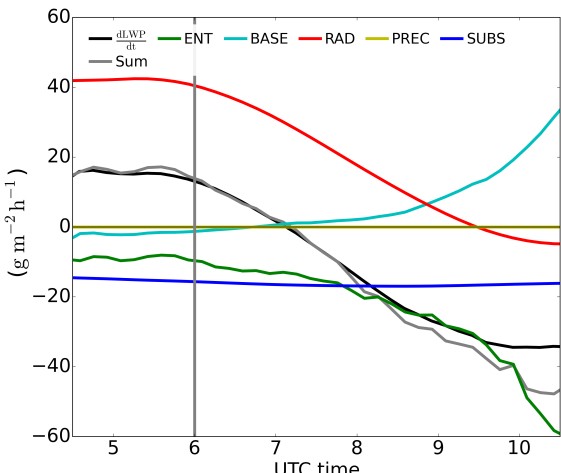

**Figure 6.** Time series of budget terms as defined in Eqs.2 and 3 with colors representing the terms as displayed in the legend, and the Sum grey line being the sum of all the terms on the RHS of Eq. 2. The vertical grey line indicates sunrise time.

### 3.3 LWP budget before and during the transition

After observing the transition in cloud characteristics and buoyancy regime, a question immediately arises: what is the relative contribution of the main physical processes driving this transition? To this end and relating to Fig. 2b, LWP is calculated and used as the metric to describe the state of the transition and calculate the budget derived by van der Dussen et al. (2014). The budget reads:

$$\frac{\partial \text{LWP}}{\partial t} = \text{BASE} + \text{ENT} + \text{PREC} + \text{RAD} + \text{SUBS} \tag{2}$$

with

$$\begin{aligned}
\text{BASE} &= \rho\eta \left( \overline{w'q'_t}^b - \Pi\gamma\overline{w'\theta'_l}^b \right) \\
\text{ENT} &= \rho w_e \left( \eta\Delta q_t - \Pi\gamma\eta\Delta\theta_l - D\Gamma_{q_l} \right) \\
\text{PREC} &= -\rho\delta P \\
\text{RAD} &= \rho\eta\gamma\delta F_{rad} \\
\text{SUBS} &= -\rho D\Gamma_{q_l} w_s(h)
\end{aligned} \tag{3}$$

with BASE representing the effect of turbulent fluxes at cloud base, ENT that of entrainment, PREC the effect of precipitation, RAD that of radiation, and SUBS the one due to subsidence. $\Delta q_t$ and $\Delta\theta_l$ are the jumps across the inversion layer for total water mixing ratio and liquid water potential temperature, respectively, defined as in van der Dussen et al. (2016): $\Delta\theta_l = \theta_l(zi^+) - \theta_l(zi^-)$ and $\Delta q_t = q_t(zi^+) - q_t(zi^-)$. $\delta P$ and $\delta F_{rad}$ represent the difference in precipitation and net radia-





tion, respectively, between the top of the inversion layer $zi^+$, assumed to be the same as Sc cloud top height in van der Dussen et al. (2016), and the Sc cloud base (van der Dussen et al., 2016). The rest of the variables in Eq. 3 are listed in Table A1.

   In short, this budget enables us to decompose the thinning or thickening of the cloud layer, quantified by a LWP tendency, and relate each contribution to the physical processes governing the stratocumulus clouds. To derive such budget van der
Dussen et al. (2014) assumed the cloud layer to be horizontally homogeneous and vertically well mixed, implying a linear increase of the liquid water with height within the cloud layer following an adiabatic liquid water profile. The first hours of the simulation perfectly fit those conditions. However, after some hours the horizontal heterogeneities created in the Sc layer and the formation of convective clouds below (see Fig. 5) make these assumptions not to longer hold. Furthermore, the assumption of one well-mixed cloud layer breaks after 10 00 UTC due to the warming by radiation and entrainment (Fig. 4). The distance
between $zi^+$ and ctop_Sc, assumed to be negligible by van der Dussen et al. (2014), increases with time up to 50 m at 10 00 UTC. For this reason we focus our analysis on the first stage of the transition until 10 00 UTC.

   Before sunrise we observe in Fig 6 a net thickening of the cloud layer by almost 20 g m$^{-2}$ h$^{-1}$, i.e. a growth of about 15%, driven solely by the longwave cooling at the cloud top (RAD term). During the entire experiment SUBS remains almost constant given the small variation of subsidence with height, showing a negative tendency of around 16 g m$^{-2}$ h$^{-1}$. The
negative tendency by entrainment (ENT) is to a large extent initially due to the entrainment of warm air (second term in ENT in Eq. 3) since, as shown in Fig.1, the free tropospheric air has similar moisture content as the cloudy air. The thinning tendency of precipitation is small, accounting for up to 4 g m$^{-2}$ h$^{-1}$ when the cloud layer its thickest. The small contribution of PREC despite large LWP is explained by the microphysical characteristics of the region. The large CCN concentrations typical for SWA (300 cm$^{-3}$ in our study) prevent any large effects of precipitation even in Sc with high liquid water content. Of similar
magnitude is the effect by cloud base fluxes before sunrise: the turbulent transport of warm air (second term of BASE in Eq. 3) dominates over its moistening effect (first term of BASE in Eq. 3) at this time. Yet the negative net effect by BASE in LWP tendency is about ten orders of magnitude smaller than that of RAD.

   After sunrise the warming effect of shortwave radiation increasingly offsets the longwave cooling at cloud top. This leads to a decreasing contribution of RAD to the thickening of the cloud layer. Due to this factor, the sign of LWP tendency changes at
around 7 15 UTC. This is the time when the thinning leading to the eventual cloud breakup starts. Correlated to the shortwave radiation increase after sunrise, the surface-driven growth of the boundary layer leads to larger entrainment rates, thus increasing the warming of the cloud layer through the free-tropospheric engulfed air. An additional factor to the already mentioned warming explains the fast shift to more negative tendencies for the ENT term after 7 00 UTC: the increased drying through entrainment. This drying increases due to two factors enhancing $\Delta q_t$, from $-0.27$ g kg$^{-1}$ at 7 00 UTC to $-1$ g kg$^{-1}$ at 10 00
UTC: the moisture input in the boundary layer by the surface; and the growth of the boundary layer itself across a drier free troposphere. This larger moisture jump enhances the impact of entrainment by a) drying the cloud layer and b) enhancing the entrainment velocity as the difference in buoyancy between the cloud and free troposphere decreases. By the end of this period, at 10 00 UTC, the positive contribution to LWP of cloud base fluxes (BASE) rises to up to 30 g m$^{-2}$ h$^{-1}$. This is explained by the increase of surface fluxes (Fig 2c) and surface buoyancy (Fig 4e) as the available net radiation at surface grows. These
changes lead to a larger contribution of the moistening $\overline{w'q_t'}^b$ term to BASE in Eq. 3, while the warming term including $\overline{w'\theta_l'}^b$




remains less variable for the first hours. Note that although the moisture flux increase at cloud base implies a growth of LWP in the budget, such growth may eventually lead to a dissipation of the cloud layer: increased surface moisture flux at surface and consequently, at cloud base, relates to enhanced buoyancy within the cloud layer, known to increase entrainment. Such accelerated entrainment leads to the warming of the upper cloud, and thus counteracts the mixing of the cloud layer necessary

for the maintenance of the Sc.

Comparing the contributions in our case before sunrise to those of the first night in van der Dussen et al. (2016), we find a RAD term almost 30% lower in our case. Given the similar LWP and $\theta_l$ jump above cloud top, we attribute the significant difference to the lack of a moisture jump here and thus, weaker cloud top radiative cooling. The BASE term reached values of about 60 g m$^{-2}$ h$^{-1}$ in van der Dussen et al. (2016), while we found very little contribution of such term during the morning

due to the compensation of moistening and warming effect of turbulent fluxes. This large difference compared to a marine case shows the relevance of the land surface, as the moistening is limited here and counteracted by a larger warming through turbulent fluxes at cloud base compared to a marine case. The nighttime ENT term is in our case about two to three times smaller than in van der Dussen et al. (2016), explained by larger turbulence created by a stronger RAD in their study. All in all, the total tendency $\frac{d\text{LWP}}{dt}$ is in the same order of magnitude for both cases although the drivers remain quite different.

The increasing negative contribution to the LWP budget by entrainment at daytime is consistent with the Sc over land case by Ghonima et al. (2016). We find our case to fall between their cases with fixed Bowen ratios of $Bo = 0.1$ and $Bo = 1$, as we observe a nearly $Bo = 0$ during night growing up to 0.6 during the day in the current case, similar to the measured conditions (Fig. 2c). This indicates the advantage of having a land surface model correctly partitioning the available net energy into surface and latent heat fluxes. The BASE term behaves in our case similar to their $Bo = 0.1$ case as it also shows a positive contribution

to cloud thickening or LWP increase.

### 3.4 Effect of wind and wind shear in the transition

#### 3.4.1 Nighttime effects

We showed that the transition from stratocumulus to cumulus over land for typical SWA conditions can take place under windless conditions. Given the recurrent presence of wind and low level jet in the morning during the observational campaign,

it is interesting to further investigate the effects that wind has on the transition. Thus, we extend the previous results considering the further effects that mean wind (MEAN) and additional wind shear at cloud top (SHEAR) have on the transition described. We include in Table 1 the timing and magnitude of the reference metrics for each experiment. Under cloudless conditions, the effect of shear at surface as well as at boundary layer top acts as a local source of Turbulent Kinetic Energy (TKE) (Conzemius and Fedorovich, 2006). In our case, such modifications in turbulence may affect the evolution of the cloud transition described

in previous sections. First, we show in Fig. 7 the relative differences between the terms defined in Eq. 3 as part of the LWP budget. Following van der Dussen et al. (2016), we show the accumulated difference, starting at 4 30 UTC, on the LWP





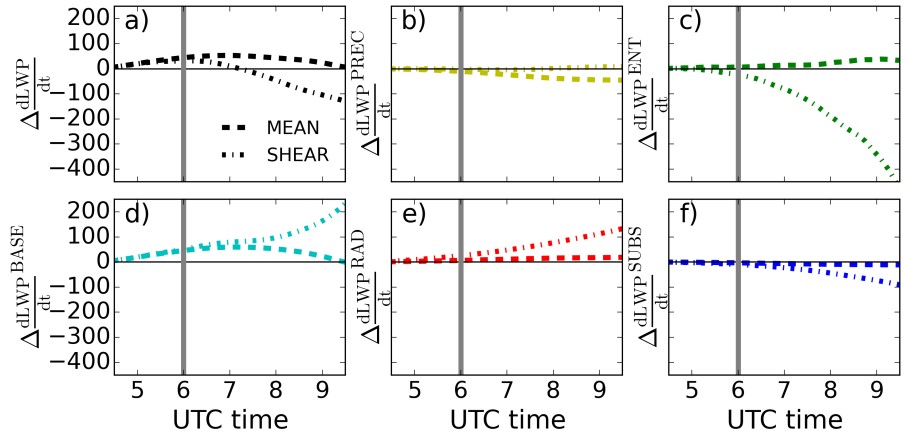

**Figure 7.** Time series of accumulated differences between MEAN and REF (dashed) and between SHEAR and REF(dotted-dashed) for each term defined in Eq. 2.

tendency due to each term between MEAN or SHEAR and the reference simulation REF. Taking the precipitation contribution PREC as an example, we calculate:

$$\Delta \frac{dLWP}{dt}^{PREC}(t) = \int\limits_{4\,30UTC}^{t} PREC(t') - PREC^{REF}(t)'dt' \tag{4}$$

and similarly for all the other terms present in Eq. 2.

The presence of a light mean wind (3 m s$^{-1}$) on the entire domain has only minor effects on the first part of the transition: Figure 7 shows a slightly larger LWP for the MEAN experiment compared to REF. The larger LWP is driven by the increased contribution of the turbulent fluxes at cloud base (BASE). For both MEAN and SHEAR it shows a thickening contribution already before sunrise, whereas it was a net thinning contribution in REF experiment. The change in BASE is explained as follows: wind enhances latent heat flux as well as turbulent generation near surface, favoring the transport of moisture to the

cloud layer. The enhanced turbulence generation near surface due to the wind, both in MEAN and SHEAR, is visible in the lower part of Fig. 8a. We show there the contributions by the buoyancy and shear terms, $B$ and $S$ respectively, to the TKE tendency budget and the good agreement on surface TKE between our experiments and the observations. Enhanced LWP in nighttime Sc by the presence of wind was also found by (Kazil et al., 2016) and attributed to enhanced buoyancy production of TKE due to latent heat release in cloud updrafts. Such findings coincide with the enhanced buoyancy term $B$ for MEAN in

Fig. 8a. Precipitation, acting as a negative feedback on LWP, attenuates the effect by BASE in the total tendency of LWP. The remaining terms show little variation between REF and MEAN.

    Wind shear at the top of the cloud layer introduces larger changes: it is known to enhance TKE locally but with a total negative effect on cloud TKE due to reduced buoyancy production (Wang et al., 2012) and to enhance entrainment at cloud top (Mellado, 2017). Before sunrise, cloud layer LWP as well as cloud base and cloud top heights (Fig. 9d) show small differences





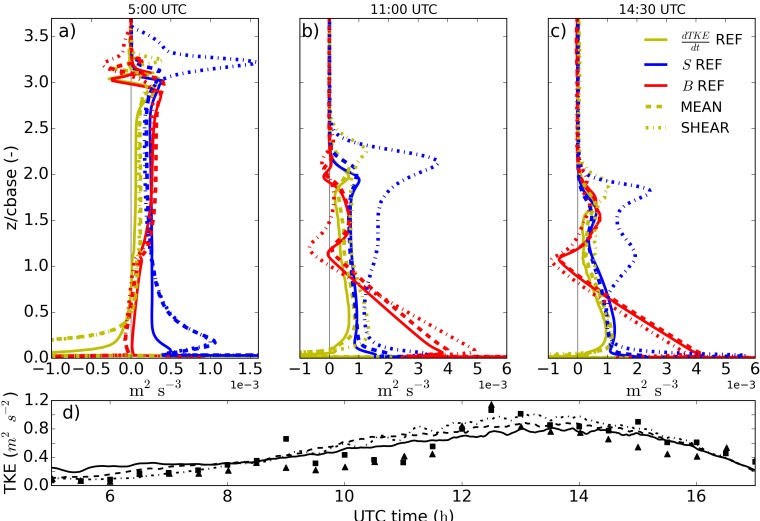

**Figure 8.** Slab averaged vertical profiles of 20-min averaged turbulent kinetic energy tendency (yellow), shear contribution (blue) and buoyancy contribution(red) for REF (full), MEAN (dashed) and SHEAR (dotted-dashed) at 5 00 (a), 11 00 (b) and 14 30 UTC(c). The height is normalized by cloud base height at the vertical axis. In (d) and following the same line settings, time series of the simulated turbulent kinetic energy at 10 m high and, in triangles and squares, as observed by two independent stations at the Savé supersite on $26^{th}$ June 2016.

between SHEAR and MEAN experiments. SHEAR shows systematic lower LWP (not shown) but a thicker Sc cloud layer, e.g. $\simeq 40$ m thicker before sunrise, due to increased entrainment velocities. Similarly, we also find a turbulent and clear sublayer between the cloud top and the inversion layer top in SHEAR (Fig. 9a). These results agree with the findings by Wang et al. (2008) and McMichael et al. (2019), who studied cloud-top shear effects on marine Sc clouds. Such agreement reinforces the analogy between the night-time Sc cloud studied here before sunrise and the typical marine Sc, given the low values of the surface fluxes.

Although with similar tendencies in the LWP budget before sunrise, the sources for turbulence and, thus, mixing within the cloudy layer are different in MEAN and REF compared to SHEAR. As shown in Fig. 8a, SHEAR shows a much larger contribution by wind shear $S$ to the TKE tendency at cloud top, up to 1.5 m$^2$ s$^{-3}$ or more than five times the local buoyancy contribution $B$ within the cloud layer. SHEAR also exhibits a slightly lower contribution by buoyancy from cloud top to surface. The larger contribution by $S$ is a consequence of the varying wind speed in the cloud boundary, while the cause for lower $B$ throughout the whole layer lies in the weaker cooling at cloud top (not shown) due to the shear-induced broader inversion layer (Mellado, 2017): the inversion layer is more than 80 m thick before sunrise at SHEAR, while is about 40 m in REF and MEAN. The increase in the depth of this layer results in a decrease of the longwave cooling at cloud top, from about $-6.1$ K h$^{-1}$ in MEAN or REF to $-4$ K h$^{-1}$ in SHEAR, as the gradients are smoothened and the time air is exposed to the





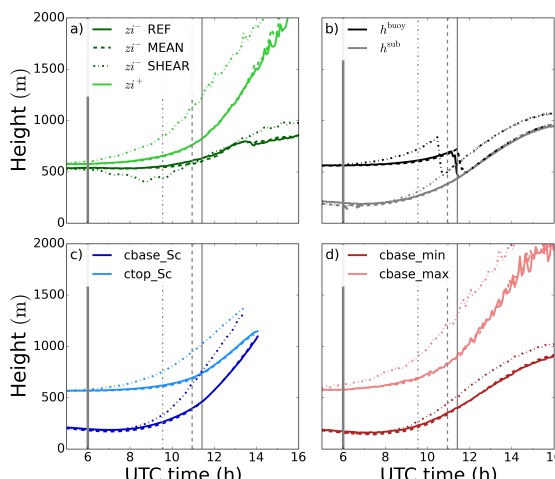

**Figure 9.** As in Fig. 5 for REF (full), MEAN (dashed) and SHEAR(dotted-dashed).

cooling is decreased (Yamaguchi and Randall, 2008). Wang et al. (2008, 2012) also found weaker cooling at cloud top and a thicker inversion layer on sheared Sc.

Our findings thus suggest that while mean wind during the night has no major effects, nighttime cloud top shear hampers the cloud growth by reducing the cooling allowed at the cloud top. The little differences between MEAN and REF reinforce the idea that the turbulence generated by wind shear at surface in MEAN needs to be transported up to the top of the well mixed layer to affect entrainment and the overall dynamics of the boundary layer. Yet the traveling turbulence is subject along its rise from surface to cloud top to the turbulent cascade, partly dissipating and having a reduced impact on the entrainment zone (Conzemius and Fedorovich, 2006). On the contrary, local shear at cloud top locally generates turbulence to immediately affect entrainment and boundary layer growth. Thus we find SHEAR to have larger effects in the Sc-Cu transition. We show in the coming section that the presence of shear at cloud top after sunrise promotes a faster breakup of the cloud layer.

### 3.4.2 Daytime effects

After sunrise the shear effects drive the cloud layer towards lower LWP due to enhanced entrainment of warm air (Fig. 7c). The enhanced entrainment velocity in SHEAR is more visible ($h$ growth as proxy for $w_e$ in Fig. 9b) after the decoupling of the cloud layer and surface between 9 00 and 10 00 UTC (see Table 1). We attribute the increase in $w_e$ not only to the presence of local shear at cloud top, but also to the positive feedback between surface fluxes and cloud thinning (Ghonima et al., 2016), further reinforced by wind shear in this case: the slightly lower LWP after sunrise in SHEAR enhances the turbulent fluxes both at surface (Fig. 8b,c) and cloud base (Fig. 7d). Larger daytime turbulence within the cloud layer leads locally to thinning of the inversion layer, allowing for a locally enhanced wind shear (Mellado, 2017) and, thus, further entrainment which will lead to a more negative rate for $\frac{d\text{LWP}}{dt}$ and the subsequent increase of surface fluxes. Furthermore, the accelerated growth of the boundary





layer in SHEAR leads to a larger moisture difference between the cloud layer and the air above, thus further reinforcing the negative effects of entrainment through additional drying in ENT (Fig. 7) in the tendency of cloud layer LWP. On the other hand, radiative cooling (RAD) remains a positive contribution for $\frac{d\text{LWP}}{dt}$ for longer time (see Fig. 7d). The reason is the thicker integration layer, caused by wind shear, over which RAD is evaluated. This layer ranges from cbase_Sc to inversion layer top ($zi^+$) for the budget in Eqs. 2,3. As assumed by van der Dussen et al. (2016), $zi^+$ and ctop_Sc agree quite well for narrow inversion layers such as the one during night without shear in REF (see Fig. 5) and the choice is unimportant. The agreement worsens when shear is present, as the inversion layer thickens and ctop_Sc and $zi^+$ show larger discrepancies (Fig. 9), as also shown by Wang et al. (2012). This thicker layer over which RAD is calculated explains the larger divergence in the net radiative flux between the cloud base and $zi^-$. Thus, a sensitive point for the discussion is the definition of the limits: one may wonder if the larger contribution to LWP gain of RAD in SHEAR may be an artifact of the boundaries selected for the budget in Eq.2. Using other limits at the top, such as ctop_Sc or cloud top, lead however to a worse closure of the budget. The negative LWP tendency is hampered in SHEAR by the positive contribution of BASE (Fig. 7d). The increase in BASE is explained as part of the positive feedback stated above: given the lower LWP at sunrise more shortwave radiation reaches the surface, increasing the surface fluxes, specially, the latent heat flux. Thus we deduce that the initial lower LWP in SHEAR accelerates the further thinning and eventual breakup of the cloud layer due to two factors enhancing entrainment: the direct enhancement due to local shear at cloud top, and the indirect one due to larger surface fluxes and boundary layer growth. This is represented in the LWP budget by more negative and positive values for ENT and BASE, respectively.

Figure 9 completes the analysis of wind sensitivity after the decoupling of the cloud layer. In agreement with the previous explanation, MEAN evolves similarly to REF as the wind-driven increase in surface fluxes is negligible compared to the boundary layer dynamics. Larger entrainment velocities accelerate the growth of the boundary layer in SHEAR, as well as the rise of cloud top and cloud base of both the total and Sc cloud layers (Figs. 9c,d). The faster growth of the boundary layer with shear at its top is a well documented feature (Conzemius and Fedorovich, 2006; Liu et al., 2018). The faster-rising cloud layer in SHEAR coincides with an earlier negative buoyancy flux minimum at cloud base and, thus, an earlier decoupling of the cloud layer by almost 2 hours (Table 1). Besides the shear effects the larger surface fluxes due to a lower LWP after sunrise also explain the faster growth of the subcloud layer buoyancy flux minimum. Similarly, the inversion layer grows faster in SHEAR due to both a decreasing $zi^-$ and a growing $zi^+$. Following the accelerated processes in SHEAR, the breakup of the cloud layer takes place about 2 hours earlier than in REF.

## 3.5 Representation by large scale metrics

A transition from stratocumulus to shallow cumulus represented as a continuum and over a period spanning several hours, such as the one shown here, poses challenges to coarser resolution models in correctly representing cloud fraction, inversion layer height or thickness as well as buoyancy source(s). To quantify the transition and explain its possible drivers beyond 10 00 UTC we calculate two metrics traditionally used in larger scale models: the ratio between subcloud layer top buoyancy flux, i.e. the buoyancy flux evaluated at $h^{\text{sub}}$, and surface buoyancy flux $r_{\theta_v} = \frac{\overline{w'\theta'_v}^{sub}}{\overline{w'\theta'_v}^{s}}$ and its analogous for the total moisture flux





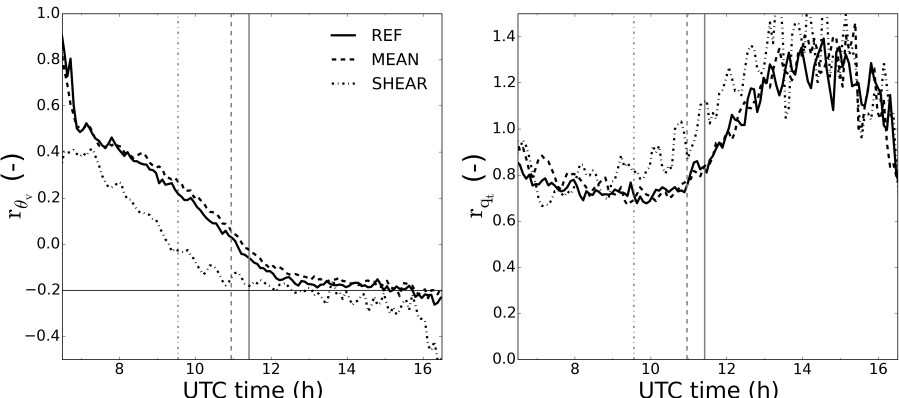

**Figure 10.** Time series of subcloud layer top to surface buoyancy flux (left) and total moisture flux (right) ratio for REF (full lines), MEAN (dashed) and SHEAR (dotted-dashed). Vertical lines represent cloud breakup for each experiment.

$r_{q_t} = \frac{\overline{w'q_t'}^{sub}}{\overline{w'q_t'}^s}$ in Fig. 10. $r_{\theta_v}$ is frequently used to parameterize entrainment velocities at boundary layer top, while $r_{q_t}$ provides information on the transport of moisture from surface to cloud and subcloud layer. These two metrics show the impact of the surface fluxes, in terms of buoyancy and moisture, on the boundary layer and capture the dynamics of it.

$r_{\theta_v}$ shows a linearly decreasing trend, showing the lowering transport of buoyancy to the cloud layer as the cloud-top driven

circulation weakens, the surface buoyancy flux grows and so does the slope of the linearly decreasing vertical buoyancy flux. Around 11 00 UTC the sign of $r_{\theta_v}$ reverses (see also nearly negative buoyancy flux minimum at $h^{sub}$ in Fig. 4d). This explicitly indicates the decoupling between the cloud layer and the surface. After 12 00 UTC $r_{\theta_v}$ approaches the typical ratio of -0.2 for dry CBLs (Stull, 1988) until the decay of turbulence generation at surface by the end of the experiment. Similarly, $r_{q_t}$ presents slightly decreasing values from 0.8 to 0.7 after sunrise. This was also found in other studies of marine Sc by de Roode

et al. (2016), mentioning that $r_{q_t} < 1$ implies a net moistening of the subcloud layer. As shown in Fig. 6, the moistening and warming of the cloud layer by turbulent fluxes from the surface almost offsets each other in terms of LWP impact on for the first hours after sunrise. After the shift in $h$ and before the breakup at 11 30 UTC we observe growing values for $r_{q_t}$ related to increasing latent heat flux at surface. After 12 00 UTC we find values higher than 1, indicating a net drying of the subcloud layer and consequent moistening of the cloud layer by surface evapotranspiration. MEAN shows little variation from REF,

reinforcing the small effect of the mean wind in the transition. The only remarkable difference is a one hour delay in reaching values of $r_{\theta_v}$ near typical CBL of $-0.2$, due to the fact under convective conditions that mean wind may hamper the turbulent updrafts from the surface to the boundary layer top, thus reducing the related entrainment (Liu et al., 2018). SHEAR shows a qualitatively similar pattern to REF after sunrise with an earlier shift on the sign of $r_{\theta_v}$ of about two hours. Afterwards, $r_{\theta_v}$ reaches values lower than $-0.2$ in SHEAR. This suggests that as found by Conzemius and Fedorovich (2006), the buoyancy

entrainment flux is enhanced compared to clear CBLs.





**Table 1.** Values and time of the main features in the stratocumulus to cumulus transition for the three experiments REF, MEAN and SHEAR.

| **Experiment** | Max LWP (g m$^{-2}$) | Start of convective phase | $\frac{d\,\text{LWP}}{dt} < 0$ time | Decoupling time $\overline{(w'\theta_v')}^{sub} < 0$ | $h = h^{\text{sub}}$ time | Breakup time $(cc < 1)$ | CBL time $(r_{\theta_v} < -0.15)$ |
|---|---|---|---|---|---|---|---|
| REF | 174.4 | 6 55 UTC | 7 07 UTC | 11 05 UTC | 11 08 UTC | 11 25 UTC | 13 02 UTC |
| MEAN | 185.6 | 6 51 UTC | 7 06 UTC | 11 14 UTC | 11 23 UTC | 10 58 UTC | 12 09 UTC |
| SHEAR | 173.9 | 6 53 UTC | 6 44 UTC | 9 19 UTC | 10 28 UTC | 9 33 UTC | 10 43 UTC |

## 4   Conclusions

Based on observations of the DACCIWA project in southern West Africa we designed a numerical experiment to reproduce the transition from nighttime stratocumulus to daytime cumulus clouds over land. Special emphasis is placed on the the stratocumulus deck breakup and the role of the surface and boundary-layer processes. This was done by means of a Large Eddy Simulation with an interactive radiation scheme and a plant-mechanistic land surface model, allowing for coupled responses of radiative profiles and surface fluxes to changes in the thermodynamic fields and surface conditions. Numerical experiments were evaluated against a complete set of observations.

We quantified the transition in terms of inversion layer height and thickness, cloud-top and cloud-base heights and boundary layer height. These metrics remain largely constant over time during the night and similar to typical marine stratocumulus clouds, and start diverting from these values a few hours after sunrise. The main drivers are the increased entrainment due the enhanced turbulence driven by the surface fluxes and, to a lesser extent, the shortwave radiative warming at cloud top. We further showed how temperature, vertical velocity distributions and buoyancy and radiative fluxes vary during the transition period. Notable features during the transitions are the decoupling of the cloud layer by 11 UTC supported by 1) two independent well mixed layers seen in the temperature profiles and 2) a negative subcloud buoyancy flux minimum. The radiative fluxes shift from exerting a net cooling effect to a warming within the cloud layer which, in addition to the warming by entrainment, leads to the mentioned decoupling.

We further described and quantified the varied physical processes that maintain and thin the stratocumulus cloud layer using the LWP budget (van der Dussen et al., 2016). The radiative term is the most dominant process contributing to LWP increase during nighttime, while its contribution decreases after sunrise and becomes a sink of LWP due to increasing shortwave radiation warming. Subsidence has a negative and fairly constant contribution to the budget during the whole transition. Precipitation and cloud base fluxes, the latter driven by the cloud top cooling circulation, have almost no effects during the night. As the day progresses, the moisture flux from the surface contributes increasingly to the growth of LWP. Entrainment has a negative and nearly constant contribution during night. After sunrise, the entrainment induced LWP thinning intensifies due to cloud layer rise and the increasing moisture difference between the cloud layer and the air above.

Lastly we investigated the effect of wind on the transition: two additional experiments were performed along with the windless reference experiment: one experiment with a mean wind of 3 m s$^{-1}$ at all heights and another with an additional





wind jump at cloud-top of $5$ m s$^{-1}$ and further increase of $5$ m s$^{-1}$ km$^{-1}$ above. The geostrophic wind was assumed to be identical to the prescribed wind in each experiment. The aim was to represent the main features of a recurrent low level jet observed in the region during the nighttime and morning. We found the mean wind to have almost no impact on the transition. However, the shear at cloud top had larger effects. Before sunrise, the inversion layer was thicker and the TKE generation

by shear higher at cloud top at the expense of lower generation by buoyancy. These features are typical of sheared marine Sc. After sunrise, shear accelerated cloud thinning, boundary layer growth and the transition to a convective boundary layer. This was due to the direct effect of shear on entrainment growth similar to clear convective boundary layers, but also to the enhanced surface fluxes as cloud layer thinned faster. The related enhanced entrainment contributed to a faster thinning of the cloud layer, leading to a breakup two hours earlier than the no-wind experiment.

We calculated widely-used relationships that characterize the prototypical clear and cloudy boundary layer to determine their ability in reproducing the transition. We find that the ratio between the subcloud layer entrainment and the surface turbulent buoyancy fluxes $r_{\theta_v}$ decreases linearly with time during the transition, starting from initial values of $r_{\theta_v} = 1$ and reaching typical dry convective values of $-0.2$ about one hour after the Sc deck break up at about 12 30 UTC. The analogous moisture ratio shows a slight decrease from $0.8$ to $0.7$ until the shift in buoyancy flux minimum. After the shift, $r_{q_t}$ increases reaching

values above 1, thus moistening the cloud layer. Mean wind leaves the transition representation by $r_{\theta_v}$ and $r_{q_t}$ unaffected, except for a 1 hour delay in reaching CBL values for $r_{\theta_v}$. In contrast, the presence of cloud top shear accelerates by 2 hours the evolution of both $r_{\theta_v}$ and $r_{q_t}$. Furthermore, $r_{\theta_v}$ reaches values more negative than $-0.2$ after breakup. These findings reveal the relevance of the land-atmosphere feedbacks on the stratocumulus thinning and cloud transition, and the impact of wind on it.

*Code and data availability.* The data obtained during the DACCIWA campaign at Savè supersite is available on the SEDOO database (http://baobab.sedoo.fr/DACCIWA/) (Derrien et al., 2016; Handwerker et al., 2016; Kohler et al., 2016). The DALES code is freely available for download at https://github.com/dalesteam/dales.

*Author contributions.* XPB wrote the manuscript with contributions of all co-authors. XPB made the analysis and produced the figures with contributions from BA,KB,CD,NK,FL,ML. JV and SdR contributed to setting up the case on LES. BA, KB, CD, NK, FB, ML, SdR and JV

contributed to the analysis of the results. BA, CD, NK, FB, ML and XPB conducted the ground measurements.

*Competing interests.* T

he authors declare that they have no conflict of interests.





# Appendix A: Appendix A. List of symbols

*Acknowledgements.* The authors would like to thank the work of all people involved in the measurement campaign of DACCIWA, and in particular to all the people who contributed to the Save supersite. The first author acknowledges M. Zouzua for providing the cloud cover observations, Anna-Lena Deppenmeier for providing ERA-interim data and Shantonu Abe Chatterjee for the useful tips during the writing
5   process. The research leading to these results has received funding from the European Union 7th Framework Programme (FP7/2007-2013) under Grant Agreement no. 603502 (EU project DACCIWA: Dynamics-aerosol-chemistry-cloud interactions in West Africa. The numerical simulations were performed with the supercomputer facilities at SURFsara and financially sponsored by the Netherlands Organisation for Scientific Research (NWO) Physical Science Division (project number SH-312-15). This study was supported by the grant from the NWOALW Open Programme (824.15.013).





**Table A1.** TEXT

| Variable | Name | Units | Variable | Name | Units |
|---|---|---|---|---|---|
| $B$ | Buoyancy term in TKE tendency equation | $m^2 s^{-3}$ | $\gamma$ | $\frac{\partial q_s}{\partial T}$ | $g_w\ Kg_a^{-1}\ K^{-1}$ |
| $Bo$ | Bowen ratio | (-) | $\Gamma_{q_l}$ | $g\eta\left(\frac{q_s}{R_d T} - \frac{\gamma}{c_p}\right)$ | $g_w$ |
| $cc$ | Cloud cover | (-) | $\delta F_{rad}$ | Difference in net radiation between $zi^+$ and cbase_Scu | $W\ m^{-2}$ |
| $c_p$ | dry air specific heat | $J\ kg_a\ K^{-1}$ | $\delta P$ | Difference in precipitation between $zi^+$ and cbase_Scu | $g_w\ g_a^{-1}\ m\ s^{-1}$ |
| cbase_Sc | Stratocumulus cloud base height | m | $\Delta q_t$ | $q_t$ jump along inversion layer | $g_w\ Kg_a^{-1}$ |
| ctop_Sc | Stratocumulus cloud top height | m | $\Delta \theta_l$ | $\theta_l$ jump along inversion layer | K |
| $D$ | cloud layer depth | m | $\eta$ | Thermodynamic factor (see van der Dussen et al. (2014)) | (-) |
| $F$ | Net radiative flux | $W\ m^{-2}$ | $\theta$ | Potential temperature | K |
| $F_0$ | Net radiative flux at surface | $W\ m^{-2}$ | $\theta_l$ | Liquid water potential temperature | K |
| $g$ | gravitational acceleration | $m\ s^{-2}$ | $\theta_v$ | Virtual potential temperature | K |
| $h$ | Boundary layer height | m | $\Pi$ | exner function | (-) |
| $h^{sub}$ | Subcloud layer height | m | $\rho$ | air density | $Kg_a\ m^{-3}$ |
| $LE$ | Latent heat flux | $W\ m^{-2}$ | | | |
| LWP | Liquid Water Path | $g_w m^{-2}$ | | | |
| $q_s$ | Saturation specific humidity | $g_w\ Kg_a^{-1}$ | | | |
| $q_t$ | Total specific humidity | $g_w\ Kg_a^{-1}$ | | | |
| $Rd$ | dry air gas constant | $J\ kg_a\ K^{-1}$ | | | |
| $r_\phi$ | Subcloud to surface $\overline{w'\phi'}$ ratio | (-) | | | |
| $S$ | Shear term in TKE tendency equation | $m^2 s^{-3}$ | | | |
| $SH$ | Sensible heat flux | $W\ m^{-2}$ | | | |
| $S_w$ | Skewness | (-) | | | |
| $T$ | Temperature | $K$ | | | |
| U | Horizontal windspeed | $m\ s^{-1}$ | | | |
| $w_e$ | Entrainment velocity | $m\ s^{-1}$ | | | |
| $w_{subs}$ | Subsidence | $m\ s^{-1}$ | | | |
| $\overline{w'\phi'}$ | Turbulent flux of $\Phi$ | $m\ s^{-1}[\Phi]$ | | | |
| $\overline{w'\phi'}^b$ | Turbulent flux of $\Phi$ at cloud base | $m\ s^{-1}[\Phi]$ | | | |
| $zi^+$ | Inversion layer top height | m | | | |
| $zi^+$ | Inversion layer bottom height | m | | | |



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
