# Peer review of "The diurnal stratocumulus-to-cumulus transition over land in southern West Africa"

_Atmospheric Chemistry and Physics, 2019_

## Referee Comment (RC1) · Anonymous Referee #1 · 13 Sep 2019

This well written article describes a number of large eddy simulation (LES) experiments of the diurnal stratocumulus-to-cumulus transition over land. The study compliments the DACCIWA field campaign that observed the stratocumulus to cumulus transition over West Africa, by seeking to provide a mechanistic description of the transition and to compare and contrast the mechanisms with those observed in the substantially more well studied marine stratocumulus. I think the article will be of wide interest and provides important insight into little studied and poorly understood processes. The experiments seem well posed and the analysis of results is exemplary. It is my opinion that article should be accepted for publication following revisions to address my concerns below.

While the paper is quite detailed, the article glosses over relatively important details

[Figure]

about the numerical implementation of the LES. The article does mention that DALES version 4.1 was used for the study, however that version of the code offers numerous options for how the equations of motion are discretized. It is well known that details of how numerical errors from the discretized equations of motion interacts with the subgrid-scale closures is important in determining the fidelity of the simulations (e.g. Pressel et al, 2017). It is also possible that the conclusions drawn here, could be quite sensitive to which set of discretization were used in the simulations. While numerical sensitivity tests would be nice, it is probably not realistic to ask for more simulations to be done. However, without such sensitivity tests it is imperative to provide a detailed description of the LES to make the experiments reproducible by others.

Not unrelated, the authors also fail to mention if their LES uses a Galilean transformation of the of the equations of motion. Depending on the numerical schemes used in the LES, it could well be that the simulations with mean wind and shear are highly sensitive to the assumed domain translation velocity. In particular, I am concerned that in the shear case if the Galilean transformation is such that the transformed mean wind in the stably-stratified free troposphere is significantly non-zero, a large component of the increased entrainment may be driven by numerical error (in the form of oscillations) in either the momentum or scalar fields. This would be particularly likely if the momentum equations were discretized using conservative centered difference scheme, as stable stratification would likely prevent the SGS model from providing sufficient dissipation damp the numerical oscillations inherent with such schemes. If the simulations do not use a Galilean transformation, I strongly recommend that they do, with a transformation selected to keep the transformed mean wind close to zero in the free troposphere.

Small Correction: p3 L11: rol -> role

Pressel, K. G., Mishra, S., Schneider, T., Kaul, C. M. and Tan, Z. ( 2017), Numerics and subgrid‐scale modeling in large eddy simulations of stratocumulus clouds, J. Adv. Model. Earth Syst., 9, 1342– 1365, doi:10.1002/2016MS000778.

---

## Referee Comment (RC2) · Anonymous Referee #2 · 27 Sep 2019

Suggestion: Minor Revision

Summary: This manuscript presents a set of large-eddy simulations (LES) of diurnal stratocumulus-to-cumulus (Sc-to-Cu) transition over southern West Africa during its monsoon season. It complements well the existing literature on the Sc-to-Cu transition over other continental regions and subtropical oceans. Specifically, it highlights the important roles of (1) the strong daytime land surface heat and moisture fluxes and (2) the wind shear by monsoonal lower-level jet on the boundary layer growth and decoupling that lead to Sc break-up. I think this manuscript is generally well-written and the analyses are comprehensive. Thus, I have only some specific comments on further clarifying the interpretation of the results. My detailed comments are as follows.

Specific Comments:

[Figure]

Title: This title is too broad. I would suggest making it more specific, e.g., 'the diurnal stratocumulus-to-cumulus transition over land during the southern West Africa monsoon season'.

Page 2, L18: Although I agree that the Sc-to-Cu transition is important for quantifying the Sc radiative effect and its bias in climate models, it would also be nice to discuss briefly why the formation of Sc after sunset is less important. Do all climate models simulate the Sc formation correctly? The Sc-to-Cu transition would be irrelevant to climate models if they cannot even produce Sc in the first place.

Page 3, L9: Could the authors please briefly summarize the mechanisms and processes by Ghonima et al. (2016), since the readers may not be familiar with them?

Page 3, L13: It would be better to clarify that the longwave and shortwave radiation has different effects on the maintenance of Sc.

Page 3, L22: It seems to me that the entrainment velocity should increase, not decrease when cloud-top wind shear is present.

Page 4, L11: The land model is not sufficiently justified. Is the surface homogeneous, and is it free of topography? Why can these assumptions be made?

Page 4, L14: What vegetation is present in SWA? Are the 2 big-leaf scheme parameters tuned specifically for the regional vegetation, or for the mix of grass and bushes (or corn) at the surface flux observation sites (see Page 5, L14 and 19)? How sensitive is the model to the choice of land/vegetation scheme?

Page 4, L20: By 'other chemical compounds', did the authors mean only the radiatively active gases, or are aerosols also included? If so, how are the aerosol concentrations prescribed?

Page 5, L5: How should I interpret the observed cloud base height: is it a local value, the domain average, or the minimum? This is especially important after the continuous Sc-deck breaks up (e.g., the sudden jump of the red circle from 1000m to 500m at 12

00 UTC in Figure 2b requires further clarification).

Page 5, L16: What does 'sonic temperature and humidity measurements from fast infrared hygrometer' mean: is it a sonic or optical equipment?

Page 5, L19: What is the motivation of using an additional site over corn for TKE measurements but not for surface fluxes?

Page 6, L5: The word 'coupled' is confusing. Did the authors mean the atmosphere and land surface are coupled, or the cloud layer and the surface air layer are coupled?

Page 6, L15: The lower level divergence is about $8 \times 10^{-6}$ s$^{-1}$, which is even stronger than the typical conditions for marine Sc (e.g., DYCOMS). Also, the subsidence profile is very shallow with a scale height of only 300m. How are these values chosen? Are they selected to keep a steady Sc-deck during nighttime? Although the authors stated that the COSMO and ERA-I both show a large spread of subsidence, it would still be necessary to demonstrate that the prescribed subsidence falls within the ranges of COSMO and ERA-I, and that the subsidence profile does not change during the diurnal cycle. It would also be necessary to discuss briefly the model sensitivity to the prescribed subsidence.

Page 6, L22: Is radiation calculated column-by-column, accounting for the spatial inhomogeneity in cloudiness?

Page 6, L31: 'Drying' is a process that makes something drier (e.g., entrainment drying), but I think the authors may instead mean that the air mass above cloud top is drier than below.

Page 7, L7: Do the free-tropospheric temperature and moisture profiles drift?

Page 7, L27: With no large-scale wind, does the surface flux rely entirely on the surface wind produced by turbulent motion in the LES (without additional gust)? How much is the LES surface wind changed by imposing the 3m/s horizontal wind in the MEAN case?

Figure 2: On L5 of the caption, the word 'red circles' should be changed to 'red trian-gles' based on panel (c).

Page 8, L15: It should be clarified that the sudden jump in surface fluxes is only shown in observations, whereas the change in surface fluxes is very smooth. For LES, most of the surface flux increase occurs well before the cloud break up, and the correlation between surface flux and cloud fraction seems very weak in the LES. However, there is a clear negative correlation between LWP and surface fluxes, which is not discussed.

Page 9, L1: Following the comment above, the statement that 'the surface fluxes are radiation-driven' is not well supported by the presented data: (1) the sudden change does not occur in LES, and (2) the coincidence between the sudden jumps in surface fluxes and cloud cover does not imply causality (or which one drives the other), and (3) cloud cover is not a good proxy for the cloud radiative effect, as the clouds can thin significantly while maintaining 100% cover. I would suggest the authors add a panel in Figure 2 to show the surface insolation, or fraction of insolation reaching the surface (both LES and observation if available). The statement would be better supported if the insolation jump occurs earlier than the surface flux jump.

Page 9, L5: In LES, the cloud cover decreases quasi-linearly only after 13 00 UTC, i.e., about 1.5 hours later than the initial break-up.

Figure 3: I suggest adding a panel showing the vertical profiles of domain-mean cloud fraction and liquid water content for completeness.

Page 10, L2: I suggest using 'thin' instead of 'narrow' because the inversion layer extends vertically, not laterally.

Page 10, L4: It may be worth clarifying that the 'inversion layer' after decoupling in-cludes the entire conditionally unstable Cu-layer, and is much thicker than commonly known sharper inversion layer that tops the Cu-layer (at around 1200 m in Figure 3e). Also, how sensitive is the definition of $z_i+$ and $z_i-$ to the threshold of 5%? It seems that

a threshold of 15% to 20% would identify the aforementioned sharper inversion layer.

Page 11, L13: It was unclear to me initially that the authors are already talking about the conditions at 14 30 UTC ('convective clouds above 950m'), so I suggest adding some reference to the time. I would also suggest moving the next sentence (about 11 00 UTC conditions) in front of this sentence based on the timeline.

Page 15, L3: Is there a reference for the statement that enhanced buoyancy within the cloud layer (instead of near the cloud top) increases entrainment?

Figure 7: Since the plotted time-series represent the time-accumulated differences, I would suggest removing the 1/dt from the y-label to avoid confusion.

Page 16, L7: I suggest clarifying that the BASE contribution is from the increased moisture flux (rather than sensible heat flux, which is almost zero at nighttime).

Figure 8: To better support the discussion (Page 17, L9), I suggest adding horizontal lines that indicate cloud top heights in all panels.

Page 17, L15: I suggest adding the reference to Figure 9a for better clarity.

Figure 9(d): Should the legend 'cbase_max' be 'ctop_max' instead?

Page 18, L3: The statement that SHEAR 'hampers the cloud growth' is not well supported by the figures: the differences in cloud top and base heights are insignificant in Figure 9(c), and the max cloud top height is even higher in SHEAR than the other cases in Figure 9(d). Could the authors provide further clarifications?

Page 18, L4: The statement that MEAN and REF differ little seems inconsistent with Figure 7, where dLWP seems similar for MEAN and SHEAR, and they both produce larger LWP than REF. Could the authors provide further clarifications?

Page 18, L9: As this subsection is focused on nighttime effects, the statement that 'SHEAR has larger effects in the Sc-Cu transition' seems a bit irrelevant, because the transition occurs several hours after sunrise. This statement may be more appropriate

for the next subsection (daytime effects).

Page 18, L12: The first paragraph of Section 3.4.2 discusses various different effects and is a bit too long to read. I suggest breaking it up into shorter paragraphs.

Page 18, L19: Based on Figure 9(a), the daytime inversion layer seems to be thickening, not thinning.

Page 20, L13: The latent heat flux at the surface appears as the denominator of the r_qt formula (Page 20, L1). Why does its increase imply a larger, not smaller r_qt?

Page 21, L10: I would suggest adding a brief summary of the distinct features of the SWA Sc clouds from the typical marine Sc (e.g., lower cloud top but higher LWP).

Typographical Comments:

Page 3, L11: No hyphen is needed for 'subtropical'. Also, the word 'role' is misspelled as 'rol'.

Page 3, L17: There is a redundant space.

Page 3, L19: The letter 'd' at the end of the sentence seems redundant.

Page 5, L15: 'Ultrasonic' should be one word.

Page 16, L13: The author 'Kazil et al.' should be placed outside the bracket.

Page 19, L3: The reference to Fig 7(d) should be Fig 7(e) instead.

Page 24: Please be consistent on the capitalization of names.

---

## Author Comment (AC1) · 12 Nov 2019

Dear co-editor:

We acknowledge the comments by the two referees. We have answered to their concerns point by point below these lines, by providing the reviewer comments in black and our response in blue. At the end of the responses we provide the revised manuscript with additions in blue and removed parts in red. Note also that some figures have been modified following referees´ comments.

Kind regards,

Xabier Pedruzo-Bagazgoitia

**Response to RC1**

This well written article describes a number of large eddy simulation (LES) experiments of the diurnal stratocumulus-to-cumulus transition over land. The study compliments the DACCIWA field campaign that observed the stratocumulus to cumulus transition over West Africa, by seeking to provide a mechanistic description of the transition and to compare and contrast the mechanisms with those observed in the substantially more well studied marine stratocumulus. I think the article will be of wide interest and provides important insight into little studied and poorly understood processes. The experiments seem well posed and the analysis of results is exemplary. It is my opinion that article should be accepted for publication following revisions to address my concerns below.

We thank the reviewer for her/his time and comments, and answer to each of the comments one by one below.

While the paper is quite detailed, the article glosses over relatively important details about the numerical implementation of the LES. The article does mention that DALES version 4.1 was used for the study, however that version of the code offers numerous options for how the equations of motion are discretized. It is well known that details of how numerical errors from the discretized equations of motion interacts with the subgrid-scale closures is important in determining the fidelity of the simulations (e.g. Pressel et al, 2017). It is also possible that the conclusions drawn here, could be quite sensitive to which set of discretization were used in the simulations. While numerical sensitivity tests would be nice, it is probably not realistic to ask for more simulations to be done. However, without such sensitivity tests it is imperative to provide a detailed description of the LES to make the experiments reproducible by others.

We added the following information in Page 5, Lines 4-8:

"The subgrid turbulence is parameterized using a TKE model following Deardorff (1980). $5^{th}$ and $2^{nd}$ order schemes are used to compute the advection over horizontal and vertical directions, respectively. The integration of the governing equations over time is carried out using a third-order Runge-Kutta scheme. The complete information on these numerical aspects is described in Heus et. Al (2010). In particular, equations 25, 43 and 49."

Not unrelated, the authors also fail to mention if their LES uses a Galilean transformation of the of the equations of motion. Depending on the numerical schemes used in the LES, it could well be that the simulations with mean wind and shear are highly sensitive to the assumed domain translation velocity. In particular, I am concerned that in the shear case if the Galilean transformation is such that the transformed mean wind in the stably-stratified free troposphere is significantly non-zero, a large component of the increased entrainment may be driven by numerical error (in the form of oscillations) in either the momentum or scalar fields. This would be particularly likely if the momentum equations were discretized using conservative centered difference scheme, as stable stratification would likely prevent the SGS model from providing sufficient dissipation damp the numerical oscillations

[Figure]

Figure 1.. Liquid water path budget for SHEAR (full lines) and SHEAR_GAL (dashed lines) experiments. Each term is described in Eqs. 2 and 3 in the manuscript.

inherent with such schemes. If the simulations do not use a Galilean transformation, I strongly recommend that they do, with a transformation selected to keep the transformed mean wind close to zero in the free troposphere.

-We acknowledge the point raised by the reviewer. Our original simulations did not prescribe any Galilean transformation. Following his/her, we performed additional MEAN ad SHEAR simulations with Galilean transformation. More specifically, we applied a transformation of 3 m/s identical to the mean wind in MEAN_GAL, and equal 6 m/s to in SHEAR_GAL. We found no significant differences in the results in MEAN nor in the SHEAR cases. We show here the results for SHEAR (full lines) and SHEAR_GAL(dashed lines) for some of the variables since, as suggested by the reviewer, this experiment seemed the most likely to be affected by the mentioned setting. Green lines in Figure 1 in this document show that the entrainment contribution to the total LWP budget does not change

[Figure]

Figure 2. a)Vertical profile of slab net radiative flux normalized over the surface value at 5 00 UTC (red), 11 00 UTC (dark yellow) and 1430 UTC (purple). On the right (b) and following the same color code, slab averaged buoyancy flux. Full and dashed lines show the SHEAR and SHEAR_GAL experiment, respectively.

[Figure]

*Figure 3. Time series of the experiment characteristic heights as defined in Fig 5 in the manuscript. Full and dashed lines sow results for SHEAR and SHEAR_GAL experiments, respectively.*

significantly and does not have any systematic bias regardless of the Galilean transformation applied. As further shown in Figures 2 and 3 the Galilean transformation has almost no overall effect in the results.

We found similar agreements comparing MEAN and MEAN_GAL (not shown). The agreement among experiments with different Galilean transformations reinforces the consistency of the results obtained in our original simulations.

To reflect the consistency of our results we added the following text at Page 8, line 15:

"To determine the dependency of the results on the Galilean transformation, we performed two extra simulations. We reproduced the MEAN experiment with an additional grid translation of 3 m/s identical to the prescribed mean wind, and the SHEAR experiment with a grid translation of 6 m/s.  These additional experiments yielded very similar results to the original ones and confirmed the independence of our numerical experiments on this condition. For the sake of simplicity all the simulations shown here have no Galilean transformation prescribed."

Small Correction: p3 L11: rol -> role

Corrected.

**Response to RC2**

Suggestion: Minor Revision

Summary: This manuscript presents a set of large-eddy simulations (LES) of diurnal stratocumulus-to-cumulus (Sc-to-Cu) transition over southern West Africa during its monsoon season. It complements well the existing literature on the Sc-to-Cu transition over other continental regions and subtropical oceans. Specifically, it highlights the important roles of (1) the strong daytime land surface heat and moisture fluxes and(2) the wind shear by monsoonal lower-level jet on the boundary layer growth and decoupling that lead to Sc break-up. I think this manuscript is generally well-written and the analyses are comprehensive. Thus, I have only some specific comments on further clarifying the interpretation of the results. My detailed comments are as follows.

We thank the anonymous reviewer for the detailed and constructive comments that helped to improve the quality of the manuscript, and provide an answer to each of them below

Specific Comments:

Title: This title is too broad. I would suggest making it more specific, e.g., 'the diurnal stratocumulus-to-cumulus transition over land during the southern West Africa monsoon season'.

We modified it to a more specific: "The diurnal stratocumulus-to-cumulus transition over land in southern West Africa"

Page 2, L18: Although I agree that the Sc-to-Cu transition is important for quantifying the Sc radiative effect and its bias in climate models, it would also be nice to discuss briefly why the formation of Sc after sunset is less important. Do all climate models simulate the Sc formation correctly? The Sc-to-Cu transition would be irrelevant to climate models if they cannot even produce Sc in the first place.

We agree with the reviewer in that little discussion is provided for the cloud formation stage in this paper. The reason is that the formation of stratocumulus is a different process to the stratocumulus to cumulus transition studied here and, consequently, it is out of the scope of this work. That is why we decided to cite some recent studies on the formation of the stratocumulus cloud layer in the region.

 Page 2, line 10 reads:

" The arrival of the cooler, but not necessarily moister, mass of air more than a 100 km inland facilitates the onset of Sc clouds over land (Adler et al., 2019; Babic et al., 2019; Dione et al., 2019). The fact that this mass of air is characterized by cloudless conditions when over the sea reveals the importance of the land and other local factors on the cloud formation and maintenance (Adler et al., 2019; Babic et al., 2019; Lohou et al., 2019). Lohou et al. (2019) extended the previous work and summarized the four phases leading from cloud formation to dissipation: stable phase, jet phase, stratus phase and convective phase"

We believe that the mentioned studies offer enough information on the cloud formation stage.

Page 3, L9: Could the authors please briefly summarize the mechanisms and pro-cesses by Ghonima et al. (2016), since the readers may not be familiar with them?

We have included a brief summary of the main processes described by Ghonima et al. (2016).  The text reads now starting in Page 3, L 7:

"They based all their cases on vertical profiles of mid-latitude marine conditions and prescribed different Bowen ratios to regulate the surface fluxes over land. They found that the Bowen ratio of the surface determines whether the surface fluxes lead to a thinning or thickening of the cloud layer. This is proved by a set of systematic experiments. Furthermore, they provided a set of Bowen ratio -dependent feedbacks highlighting the relevant role of the land: one feedback loop where the increase of sensible heat flux would increase entrainment, thinning the cloud layer, enhancing the net radiation at surface and further increasing the sensible heat flux. They provided two more feedbacks related to latent heat flux (LE): one in which its increase moistens and thickens the cloud layer, decreasing net radiation and surface and, consequently, LE; and another in which the LE increase enhances entrainment, leading to cloud thinning and a further increase of LE."

Page 3, L13: It would be better to clarify that the longwave and shortwave radiation has different effects on the maintenance of Sc.

We have modified the sentence, that reads now (P 3, L 24):

"Net longwave radiation is the source for cloud maintenance during night through cloud-top cooling and, as the day evolves, increasing shortwave radiation becomes a factor for dissipation"

Page 3, L22: It seems to me that the entrainment velocity should increase, not de-crease when cloud-top wind shear is present.

The reviewer is right. We corrected the typo and the sentence now reads:

"They similarly concluded that entrainment velocity increases, leading to a decrease in cloud liquid water content, in presence of cloud-top wind shear."

Page 4, L11: The land model is not sufficiently justified. Is the surface homogeneous, and is it free of topography? Why can these assumptions be made?

The terrain around the measuring site was relatively flat (see also Adler et al., 2019), allowing us to assume a topography-free domain. This assumption simplifies our study and allows us to focus on the local effects that can be more easily generalized to similar situations in southern West Africa. In addition, the size of the heterogeneities were rather small in the surroundings, in the order of 50-100 metes which do not lead to formation of secondary circulations (Patton et al., 2005). Unfortunately, there was no data collected on any surface characteristic of the different heterogeneity types (see also next question). Due to all these reasons we decided to keep our assumptions to a minimal level. In short, we assume that our homogeneous land-surface responds to environmental variables allowing only for surface dynamic heterogeneities driven by the presence of clouds

Page 4, L14: What vegetation is present in SWA? Are the 2 big-leaf scheme parameters tuned specifically for the regional vegetation, or for the mix of grass and bushes (or corn) at the surface flux observation sites (see Page 5, L14 and 19)? How sensitive is the model to the choice of land/vegetation scheme?

The vegetation near and around the site consisted of heterogeneous patches containing shrubs, crops or taller trees in very dense forests. Unfortunately, no detailed measurements were taken on the vegetation types and properties during the DACCIWA campaign. Thus, the LES case was designed taking into account the only information available: the surface fluxes. The plant-based land surface model allows for spatially heterogeneous values of the surface fluxes depending on environmental conditions. This is known to be essential to simulate clear (Patton 2005) and cloudy (Sikma and Vilà-Guerau de Arellano, 2019) convective boundary layers". On the sensitivity of the model to vegetation type, Vilà-Guerau

de Arellano et al. (2014) showed that using C3 or C4 grass can impact the amount of moisture and clouds on a shallow cumulus day.

Page 4, L20: By 'other chemical compounds', did the authors mean only the radiatively active gases, or are aerosols also included? If so, how are the aerosol concentrations prescribed?

Indeed, we meant only the radiatively active gases. We did not account for any radiative effect of aerosols. We have added the following text in Page 7, Line 14:

"No radiative effects of aerosols are taken into account here."

Page 5, L5: How should I interpret the observed cloud base height: is it a local value, the domain average, or the minimum? This is especially important after the continuous Sc-deck breaks up (e.g., the sudden jump of the red circle from 1000m to 500m at 12 00 UTC in Figure 2b requires further clarification).

The observed cloud base heights are local measurements obtained above the ceilometer at a 1min resolution. The used ceilometer provided up to 3 cloud base heights for each measurement. We decided to use only the lowest one, as higher cloud detections may not necessarily relate to a cloud base but to a cloud edge, for example.

To clarify this we added the following text at Page 5, L 19:
"From the backscatter profiles three cloud base heights are obtained using the manufacturer algorithm. We select the lowest one to ensure that the detection reflects a cloud base and not, for example, a cloud edge."

Related to this, the jump mentioned by the reviewer from 1000m to 500m can be attributed to the first appearance of cumulus clouds at 500m after the rise of the stratocumulus cloud base up to 1000m or the breakup of the deck itself. We added the following text starting at Page 9, L 13:

"Therefore, the jump in cloud base height from about 1000 m to 500 m is due to either the appearance of the first shallow cumulus at 500 m after the stratocumulus cloud base rise up to 1000 m, or the breakup of the stratocumulus deck leading to different observed cloud base heights."

Page 5, L16: What does 'sonic temperature and humidity measurements from fast infrared hygrometer' mean: is it a sonic or optical equipment?

We have rewritten the text to be more clear. It reads now:

The 30-min sensible and latent heat fluxes are calculated from high-frequency (20 Hz sampling rate) measurements of wind speed and sonic temperature obtained by ultrasonic anemometer, and humidity measurements which are based on the absorption of near-infrared radiation and obtained by fast-response LI-COR sensor by applying eddy-correlation method.

Page 5, L19: What is the motivation of using an additional site over corn for TKE measurements but not for surface fluxes?

The station used for both TKE and surface fluxes was placed over a mixed of grass and bushes, while the one used only for TKE was placed over corn. The former location was more representative of the area and, thus, we used the surface fluxes there as a reference for our LES case. The use of both station observations of TKE allows us to check with higher confidence that the LES case simulated calculates the right amount of turbulence even at the lowest levels.

Page 6, L5: The word 'coupled' is confusing. Did the authors mean the atmosphere and land surface are coupled, or the cloud layer and the surface air layer are coupled?

We refer to the classification carried out by Lohou et al. (2019) in their work where they classified the observed night-day transitions during the campaign in three scenarios, the first of which is called "coupled case". Their paper offers an in-depth explanation of each case. This sentence was meant to facilitate the interpretation of the LES study as a particular case of the wider observation based-scenarios presented by Lohou et al. (2019). We have reformulated the sentence, that reads now:
"In particular, we study the Sc-Cu transition of a coupled case or Scenario 1 as described in Lohou et al. (2019)"

Page 6, L15: The lower level divergence is about 8x10ˆ-6 sˆ-1, which is even stronger than the typical conditions for marine Sc (e.g., DYCOMS). Also, the subsidence profile is very shallow with a scale height of only 300m. How are these values chosen? Are they selected to keep a steady Sc-deck during night time? Although the authors stated that the COSMO and ERA-I both show a large spread of subsidence, it would still be necessary to demonstrate that the prescribed subsidence falls within the ranges of COSMO and ERA-I, and that the subsidence profile does not change during the diurnal cycle. It would also be necessary to discuss briefly the model sensitivity to the prescribed subsidence.

As already guessed by the reviewer, the subsidence profile was designed within the COSMO and ERA profiles and such that the cloud top height would be constant during the night time. For simplicity we assumed a constant subsidence on time during the simulation, although we are aware that such assumption cannot be validated given the lack of observations and the uncertainty among models. The study of the transition under different or changing subsidence profiles with time is a whole study in itself (see van der Dussen et al. 2016) and, therefore, out of the scope of the present study.

We have modified the related part of the text, that reads now (P 6, L 22):

"Our choice for the subsidence profile was based on a cloud top equilibrium between subsidence motions and net longwave cooling. In doing so, our main purpose is to mimic and reproduce the physics of nocturnal stratocumulus clouds that are characterized by constant cloud top height during the night. This is justified given the uncertainty and high temporal variability in subsidence profiles, as well as its large spread among regional simulations carried out with the Consortium for Small-Scale Modeling (COSMO) within the DACCIWA project or ERA-interim reanalysis. For simplicity we assume the subsidence profile to be constant on time during the whole simulation."

Page 6, L22: Is radiation calculated column-by-column, accounting for the spatial inhomogeneity in cloudiness?

That is correct. To clarify it we added the following sentence when describing the radiative scheme at Page 4, L 28:
"This scheme allows to represent the surface dynamic heterogeneities caused by cloud spatial inhomogeneities as it provides a column-dependent net radiation at the surface."

Page 6, L31: 'Drying' is a process that makes something drier (e.g., entrainment drying), but I think the authors may instead mean that the air mass above cloud top is drier than below.

That is correct. We have re-written the sentences to clarify this. It reads now:

" Subtropical marine Sc clouds are frequently capped by much drier air above cloud top (Duynkerke et al. ,2004 and Wood, 2012). Yet none of the radiosonde profiles show any strong jump in moisture above 570 meters."

Page 7, L7: Do the free-tropospheric temperature and moisture profiles drift?

The reviewer is right in pointing out that the presence of subsidence will affect the lapse rates along the day. However, the first order effect of subsidence is to change the absolute value of the quantity that is being advected vertically, and to a much lesser extent it modifies the lapse rate. The thermodynamic impact of the boundary layer growth and its related dynamics are orders of magnitudes stronger up to the boundary layer height (around 1000 m). Above the maximum boundary layer height, the subsidence-driven net drying and warming during the entire simulation is less than 0.6 g/kg and 0.8 K respectively.

If the reviewer referred to any effect of horizontal advection, we did not prescribe any horizontal advection of moisture, temperature or any other scalar. This was due to our intention of keeping a minimal-element transition focusing on local effects, and given the lack of reliable observations on advection.

Page 7, L27: With no large-scale wind, does the surface flux rely entirely on the surface wind produced by turbulent motion in the LES (without additional gust)? How much is the LES surface wind changed by imposing the 3m/s horizontal wind in the MEAN case?

[Figure]

*Figure 4.Vertical profiles of slab average wind for REF and MEAN.*

The surface fluxes in the REF experiment are resolved by the land surface model and are dependent on near surface wind as well as soil moisture, air $CO_2$ concentration, temperature and humidity and available direct and diffuse radiation. In absence of large-scale wind, the wind will indeed be mostly due to the turbulent motions and the organized downdrafts and updrafts within the boundary layer, with its related divergence and convergence zones at the surface. Figure 1 shows the slab average horizontal wind for REF and MEAN and that near the surface the actual difference is of between 1 and 1.5

m/s. The slab average values for REF are nearly zero, although locally there can be values of up to 2.5 m/s.

Figure 2: On L5 of the caption, the word 'red circles' should be changed to 'red trian-gles' based on panel (c).

Corrected.

Page 8, L15: It should be clarified that the sudden jump in surface fluxes is only shown in observations, whereas the change in surface fluxes is very smooth. For LES, most of the surface flux increase occurs well before the cloud break up, and the correlation between surface flux and cloud fraction seems very weak in the LES. However, there is a clear negative correlation between LWP and surface fluxes, which is not discussed.

We do not recommend doing a one-to-one comparison between observations and LES in Figure 2 in the manuscript. While the LES results are domain averaged values and representative of the boundary-layer scale, the observations are one-point measurements and have a significantly smaller footprint. As such, the observations exhibit a much larger variability than the LES and are only meant as an indicator that the values obtained through the LES experiment fall within typical observed values. One could argue that the gradual increase in surface fluxes before break up present in LES is also visible in the trend of the observations between 6 and 11 30 UTC, and that given enough point observations, their average would yield SH and LE curves similar to those of LES.

To clarify that the jump in observed surface fluxes coincides with the time at which LES predicts a cloud break up, we re-wrote the following sentence starting at Page 9 , L 7: "The breakup in the cloud layer, defined as the time when cloud cover (cc) is below 1, takes place at around 11 30 UTC in the LES experiment and coincides with the observed sharp increase in surface fluxes"

The correlation and causality between LWP and the surface fluxes is implicitly discussed and analysed in equations 2 and 3 and in Figure 6 in the manuscript. In fact, the term BASE contributing to the total LWP tendency is partly driven by the surface fluxes. We agree with the reviewer in that perhaps there was not enough information on the relation between the decrease in LWP and the increase in surface fluxes at this stage of the manuscript. We therefore modified the following sentence in Page 9, L 1: "Between 2 to 3 hours after sunrise (6 00 UTC) the cloud layer begins to rise and subsequently decreases its liquid water content (Fig. 2b), allowing more radiation to reach the surface and enhancing the surface fluxes (see Rnet, LE and SH increase between 6 and 10 UTC in Fig. 2c)."

Page 9, L1: Following the comment above, the statement that 'the surface fluxes are radiation-driven' is not well supported by the presented data: (1) the sudden change does not occur in LES, and (2) the coincidence between the sudden jumps in surface fluxes and cloud cover does not imply causality (or which one drives the other), and(3) cloud cover is not a good proxy for the cloud radiative effect, as the clouds can thin significantly while maintaining 100% cover. I would suggest the authors add a panel in Figure 2 to show the surface insolation, or fraction of insolation reaching the surface(both LES and observation if available). The statement would be better supported if the insolation jump occurs earlier than the surface flux jump.

In that sentence we only referred to the surface flux observations. We added the observed net radiation Rnet to the figure. There, the jump in Rnet from about 300 to almost 700 W/m2 within the same 30 minutes at which the surface fluxes abruptly increase demonstrates that the surface fluxes are indeed radiation driven. Although the available temporal resolution of observations shows an apparently simultaneous jump in net radiation

and surface fluxes within 30 minutes, we trust that the reviewer will agree with us in that this is enough to show that it is net radiation what drives the fluxes.

The text reads now (P 9, L 6):

"The breakup in the cloud layer, defined as the time when cloud cover cc is below 1, takes place at around 11 30 UTC in our LES experiment and coincides with the observed sharp increase in surface fluxes of about 150  W/m2, i.e., a threefold increase compared to before-breakup values. This sudden change coincides with the sharp increase of net radiation due to clod breakup (see Fig. 2c) and reveals that surface fluxes are radiation-driven at this stage."

Page 9, L5: In LES, the cloud cover decreases quasi-linearly only after 13 00 UTC, i.e.,about 1.5 hours later than the initial break-up.

We have modified the sentence. It reads now:

"About 90 minutes after cloud break up, cc decreases quasi-linearly until the end of the simulation"

Figure 3: I suggest adding a panel showing the vertical profiles of domain-mean cloud fraction and liquid water content for completeness.

We have added an additional panel with slab averaged vertical profiles of liquid water mixing ratio and horizontal cloud fraction, and few sentences describing the newly added figures.
The sentences read (P 11, L 2):

 "The liquid water mixing ratio $q_l$ shows in Fig. 3b a linear increase with height within the cloud layer typical of well mixed stratocumulus clouds (Duynkerke et al. 1995, Wood 2012)."

Page 11, L 10:

"Furthermore, the cloud fraction in the lower part of the cloud layer at 14 30 UTC resembles that of shallow cumulus clouds (Siebesma et al. 2003). In this case, however, a second peak in cloud fraction and larger $q_l$ reveals the presence of more clouds at around 1200-1300 m. These clouds are the remnants of the stratocumulus higher part."

Page 10, L2: I suggest using 'thin' instead of 'narrow' because the inversion layer extends vertically, not laterally.

*Narrow* has been substituted by *thin* when referring to the inversion layer along the whole text.

Page 10, L4: It may be worth clarifying that the 'inversion layer' after decoupling includes the entire conditionally unstable Cu-layer, and is much thicker than commonly known sharper inversion layer that tops the Cu-layer (at around 1200 m in Figure 3e).Also, how sensitive is the definition of zi+ and zi- to the threshold of 5%? It seems that a threshold of 15.% to 20% would identify the aforementioned sharper inversion layer.

The reviewer is right in that a less strict threshold would yield different zi+ and zi- and could, indeed, help in identifying the sharper threshold within what we defined inversion layer in Fig. 3g in the manuscript between 12000 and 1300m. We however decided to use the same threshold as previous studies (van der Dussen et al., 2016) and keep it constant along the whole simulation for the sake of consistency and easier comparison with other studies. We thought that, by doing this, the interpretation of the whole unstable cumulus layer as an expansion of the sharp stratocumulus inversion was a new and interesting way of looking at it.

We have added the following text, that together with the previous sentence reads now: "Such evolution of the inversion layer enables us to interpret the typically conditionally unstable region of the cloudy layer in convective conditions as an expanded analogue of the very sharp inversion layer in Sc clouds. Note that this layer includes the sharper inversion layer common in cumulus-topped boundary layers and present in Fig. 3 g,h,i between 1200 and 1300 m."

Page 11, L13: It was unclear to me initially that the authors are already talking about the conditions at 14 30 UTC ('convective clouds above 950m'), so I suggest adding some reference to the time. I would also suggest moving the next sentence (about 1100 UTC conditions) in front of this sentence based on the timeline.

We have rearranged the order of sentences chronologically and added more references to the time for the sake of clarity.

Page 15, L3: Is there a reference for the statement that enhanced buoyancy within the cloud layer (instead of near the cloud top) increases entrainment?

From Kazil et al. (2016) on stratocumulus over sea:

"in clouds with LWP $\gtrsim$ 50 g/m2, longwave emissions are insensitive to LWP. This leads to the general conclusion that in boundary layer growth and entrainment due to a boundary layer moistening arises by stronger production of TKE from latent heat release in cloud updrafts, rather than from enhanced longwave cooling."

Similarly, Ghonima et al. (2016) also found , as we did, enhanced surface latent heat flux to increase entrainment, which we attribute to larger buoyancy within the cloud layer.

Looking at our case, we observe than mean LWP is lower than 50 g/m2 only after 12 UTC. Given that our budget analysis is only valid until 11 UTC, we assume that our claim is correct. We added Kazil et al. (2016) and Ghonima et al. (2016) references to the text and included the relation between buoyancy and turbulence to clarify the relation between processes. The sentence reads now:

"increased surface moisture flux at surface and consequently, at cloud base, relates to enhanced buoyancy through latent heat release and larger turbulence within the cloud layer, known to increase entrainment."

Figure 7: Since the plotted time-series represent the time-accumulated differences, I would suggest removing the 1/dt from the y-label to avoid confusion.

We thank the reviewer for the suggestion. We have also modified the labels and Eq 4 in the manuscript.

Page 16, L7: I suggest clarifying that the BASE contribution is from the increased moisture flux (rather than sensible heat flux, which is almost zero at night time).

The sentence reads now:

"The larger LWP is driven by the increased contribution of the turbulent fluxes at cloud base (BASE) and, particularly the contribution of the moisture flux."

Figure 8: To better support the discussion (Page 17, L9), I suggest adding horizontal lines that indicate cloud top heights in all panels.

Horizontal lines showing cloud top height have been added to the figure.

Page 17, L15: I suggest adding the reference to Figure 9a for better clarity.

Reference has been added.

Figure 9(d): Should the legend 'cbase_max' be 'ctop_max' instead?

We thank the reviewer for noticing the typo. It has been corrected.

Page 18, L3: The statement that SHEAR 'hampers the cloud growth' is not well sup-ported by the figures: the differences in cloud top and base heights are insignificant in Figure 9(c), and the max cloud top height is even higher in SHEAR than the other cases in Figure 9(d). Could the authors provide further clarifications?

We mean that while MEAN showed an additional LWP growth before and right after sunrise, while the addition of shear at cloud top hampered that LWP growth. Although the differences in general are much smaller than those few hours after sunrise in terms of LWP or cloud top height, a lower accumulated LWP increase in SHEAR compared to MEAN is visible in Fig. 7 a in the manuscript around sunrise. Still and given the little relevance of these differences, we decided to remove the mentioned sentence.

Page 18, L4: The statement that MEAN and REF differ little seems inconsistent with Figure 7, where dLWP seems similar for MEAN and SHEAR, and they both produce larger LWP than REF. Could the authors provide further clarifications?

We were referring to the TKE and shear (*S*) and buoyancy (*B*) contributions near cloud top during the night. We rewrote the sentence that now reads at Page 19, L 7 :

"The little differences between MEAN and REF at cloud top turbulent properties (Fig. 8 a) reinforce the idea that the turbulence generated by wind shear at surface in MEAN needs to be transported up to the top of the well mixed layer to affect entrainment and the overall dynamics of the boundary layer."

Page 18, L9: As this subsection is focused on nighttime effects, the statement that 'SHEAR has larger effects in the Sc-Cu transition' seems a bit irrelevant, because the transition occurs several hours after sunrise. This statement may be more appropriate for the next subsection (daytime effects).

The original purpose of the sentence was to bridge between the two sections, but we agree with the reviewer in that the current placement is misleading and thus we have removed it.

Page 18, L12: The first paragraph of Section 3.4.2 discusses various different effects and is a bit too long to read. I suggest breaking it up into shorter paragraphs.

We have divided it into 3 paragraphs where the entrainment, radiation and cloud-base flux terms are separated.

Page 18, L19: Based on Figure 9(a), the daytime inversion layer seems to be thicken-ing, not thinning.

The slab average inversion layer thickness is indeed growing, but the presence of shear and enhanced turbulence near cloud top leads to a more wavy and irregular inversion layer allowing for regions where there may be larger wind shear in the vicinity of the cloud top, promoting further entrainment. Mellado (2017) provides a more detailed analysis of this phenomenon.

Page 20, L13: The latent heat flux at the surface appears as the denominator of ther_qt formula (Page 20, L1). Why does its increase imply a larger, not smaller r_qt?

We agree with the comment of the reviewer and have re-written the sentence, that reads now:

"After the shift in $h$ and before the breakup at 11 30 UTC we observe growing values for $r_{qt}$ representing a weaker moistening of the subcloud layer."

.Page 21, L10: I would suggest adding a brief summary of the distinct features of theSWA Sc clouds from the typical marine Sc (e.g., lower cloud top but higher LWP).

We believe that the features of stratocumulus over land may be quite variable and that one case may not be representative of a typical stratocumulus case in SWA. However, we do think that the mechanisms leading to dissipation or transition of such stratocumulus can be easier generalized when over land. Because of this reason, we decided to include in the conclusions the main processes causing the thinning and transition, but not the typical features of the stratocumulus deck as this may be more variable among cases.

Typographical Comments:

Page 3, L11: No hyphen is needed for 'subtropical'. Also, the word 'role' is misspelledas 'rol'.

It has been corrected.

Page 3, L17: There is a redundant space.

It has been corrected.

Page 3, L19: The letter 'd' at the end of the sentence seems redundant.

Indeed it was not necessary It has been deleted.

Page 5, L15: 'Ultrasonic' should be one word.

Corrected.

Page 16, L13: The author 'Kazil et al.' should be placed outside the bracket.

Corrected.

Page 19, L3: The reference to Fig 7(d) should be Fig 7(e) instead.

Corrected.

Page 24: Please be consistent on the capitalization of names.

Corrected.

[revised manuscript text omitted]

---

## Author Response (AR2)

Dear Editor,

We thank your to-the-point and supporting advice, and have answered and modified the manuscript as shown below in the point-by-point answer.

I have worked through your and your coauthors' responses to the reviewers' concerns and suggestions and I am mostly satisfied with them. You and your coauthors are to be commended for your clear and detailed efforts in this regard. There are however a number of minor issues that still need to be addressed:

(1) Reviewer 2 states that "it would also be nice to discuss briefly why the formation of Sc after sunset is less important. Do all climate models simulate the Sc formation correctly." You responded with comments regarding the formation process. I do not believe that this is what the reviewer was asking about. It seems to me that the reviewer was looking for a sentence or two describing why Sc formation after sunset is less important than that in the day, i.e., why do we care about Sc during the day and not as much as night. In your manuscript your statement does start out with "The high albedo of low Sc clouds and its …" and this offers clear links to the day. However, Sc do have other impacts at nighttime and it would be useful to include a sentence or two on this aspect of Sc.

To show a more clear motivation for the study of cloud evolution during the day instead of the night, we have included the following sentence at Page (P) 2, Line (L) 22:
"These biases are less relevant during the night due to the absent shortwave radiation, as cloud-induced variations in the longwave radiation are one order of magnitude smaller than those of shortwave radiation."

(2) Reviewer 2 states that "The land model is not sufficiently justified." I had the same concern when reading the manuscript. You provide useful details in your response to the reviewer. Please would you include some of these details in the manuscript, as this will help other readers with the same concerns as Reviewer 2 and me.

Answering this comment and the one right after, we added the following paragraph in the model settings section, starting at P 6, L 18:

"The vegetation near and around the site consisted of heterogeneous patches containing shrubs, crops or taller trees in very dense thickets with areas in the order of 50-100 meters, a size too small to lead to the formation of secondary circulations (Patton, 2005). Unfortunately, no detailed measurements were taken on the vegetation types and properties during the DACCIWA campaign. Thus, the case presented here shows spatially homogeneous soil and vegetation properties and is constrained taking into account the information available on the surface fluxes. The land surface model allows for spatially heterogeneous values of the surface fluxes depending on environmental conditions. This is known to be essential to realistically simulate clear (Patton, 2005) and cloudy (Sikma and Vilà-Guerau de Arellano, 2019) convective boundary layers. The terrain around the measuring site was relatively flat (Adler et al., 2019), allowing us to assume a topography-free domain. Thus, assuming a flat and homogeneous surface simplifies our study and permits to focus on the local effects that can be more easily generalized to similar situations in SWA. Meanwhile, the dynamic heterogeneities created guarantee a sufficiently realistic representation of the boundary layer during the day."

(3) Similarly to point (2) above, please include some of your explanation to Reviewer 2's questions about the vegetation parameters and the sensitivity to the choice of the land/vegetation scheme in your manuscript.

Please see the answer to the previous comment.

(4) The decision that you made regarding keeping the threshold for $z_i+$ and $z_i-$ the same as previous studies in order to facilitate comparisons with previous studies should also be included in the manuscript.

We added the additional text at P 10, L 20:

"The thresholds defining $zi^{-}$ and $zi^{+}$ are set constant over the whole simulation for the sake of consistency and easier comparison with previous studies"

(5) Page 16 line 6: the added references of Ghinoma et al and Kazil et al need to be placed within parentheses.

The references are now between parentheses.

(6) Reviewer 2 states that "Based on Figure 9(a), the daytime inversion layer seems to be thickening not thinning. Please incorporate a sentence or two in the manuscript from your response, as well as the reference to Mellado (2017).

The text now reads:

"Larger daytime turbulence within the cloud layer leads to a thicker slab average inversion layer. However, the presence of shear and enhanced turbulence near cloud top leads to a more wavy and irregular inversion layer. As a result, there is a local thinning of the inversion layer, allowing for a locally enhanced wind shear (Mellado, 2017) and, thus, further entrainment…"

(7) There are still a few grammatical errors within the manuscript, including within the text added in response to the reviewers. Please would you try to eliminate as many of these as possible before resubmitting your manuscript.

We have revised the full manuscript and corrected all the errors and typos found.

As stated above, these suggestions are all minor in nature and I do not think that they will take you long to address. I look forward to accepting your manuscript for publication once these issues are satisfactorily addressed.

Kind regards,
Sue
* * *
Susan C. van den Heever
Professor
Department of Atmospheric Science
Colorado State University
(970) 491-8501
https://vandenheever.atmos.colostate.edu/

[revised manuscript text omitted]